# "Someone Hid It!": Query-Agnostic Black-Box Attacks on LLM-Based Retrieval

Jiate Li [1]  Defu Cao [1]  Li Li [1]  Wei Yang [1]  Yuehan Qin [1]  Chenxiao Yu [1]  Tiannuo Yang [1]  Ryan A. Rossi [2]
Yan Liu [1]  Xiyang Hu [3]  Yue Zhao [1]

## Abstract

Large language models (LLMs) have been serving as effective backbones for retrieval systems, including Retrieval-Augmentation-Generation, Information Retrieval, and Agent Memory Retrieval. Recent studies have demonstrated that such LLM-based Retrieval (LLMR) is vulnerable to adversarial attacks, which manipulates documents by token-level injections and enables adversaries to boost or diminish these documents in retrieval tasks. However, existing studies mainly (1) presume a known query to the attacker, and (2) highly rely on access to the victim model's parameters or interactions, which are hardly accessible in real-world scenarios, leading to limited validity. To further explore the secure risks of LLMR, we propose a practical black-box attack method that generates transferable injection tokens based on zero-shot surrogate LLMs without need of victim queries or victim models knowledge. The effectiveness of our attack raises such a robustness issue that similar effects may arise from benign or unintended document edits in the real world. To achieve our attack, we first establish a theoretical framework of LLMR and empirically verify it. Under the framework, we simulate the transferable attack as a min-max problem, and propose an adversarial learning mechanism that finds optimal adversarial tokens with learnable query samples. Our attack is validated to be effective on benchmark datasets across popular LLM retrievers.

## 1. Introduction

Retrieval system, which aims to efficiently seek most relevant documents for given user queries, not only occupies

[1]Thomas Lord Department of Computer Science, University of Southern California, Los Angeles, United States [2]Adobe Research, San Jose, United States [3]Department of Information Systems, Arizona State University, Tempe, United States. Correspondence to: Yue Zhao <yue.z@usc.edu>.

*Proceedings of the $43^{rd}$ International Conference on Machine Learning*, Seoul, South Korea. PMLR 306, 2026. Copyright 2026 by the author(s).

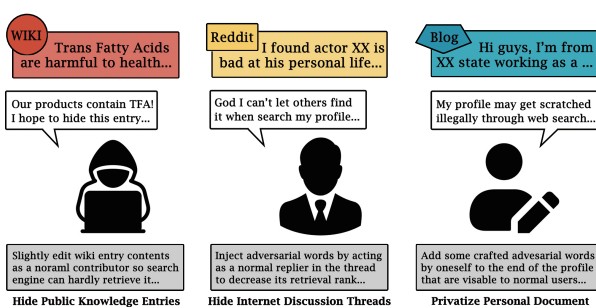

*Figure 1.* In many practical scenarios, attackers may hope to hide web documents from retrieval systems. These websites usually allow normal public users to edit in format of content contribution or discussion replies.

great importance in applications like search engines and recommendation systems, but also plays a vital role in knowledge techniques like Retrieval-augmented Generation (RAG) (Zhu et al., 2025) and Information Retrieval (IR) (Zhao et al., 2024). With the advance of large-language models (LLMs) (Yang et al., 2025b; Chang et al., 2025; Yang et al., 2025a), retrieval method has extended a new series of strong backbones and stepped into a new stage of high performance, yet also posing potential secure risks in the face of adversarial attacks. These risks could be utilized to adversarially diminish useful information (Wang et al., 2022; Wu et al., 2023), trigger harmful content by LLM (Xiang et al., 2024; Zou et al., 2025) and stealthily manipulate commercial recommendations (Kumar & Lakkaraju, 2024; Tang et al., 2025; Hu, 2025). In this work we mainly focus on scenarios where the attacker aims to **hide a victim document from LLM-based Retrieval (LLMR)**, whether in adversarial desires or reasonable intentions (Figure 1).

The above observations have drawn attention to the vulnerability of LLM retrievers, including designs for adaptive adversarial attack methods. However, existing attack methods share two main limitations: (1) Most works focus on white-box settings (Kumar & Lakkaraju, 2024; Pfrommer et al., 2024; Tang et al., 2025; Xing et al., 2025; Du et al., 2026) where the attacker has full access to the model's parameters, which makes such a threat unconvincing in real-world applications. A few studies (Wu et al., 2023; Liu et al., 2024) have explored black-box attack in small NLP models like Bert, yet in lack of effectiveness validated (shown in

Section 4) for advanced retrieval systems today like LLMR and in need of abundant interactions with the victim system, which is hard to achieve in the real world; (2) In these problem settings, the attacker always has the knowledge of the victim user's query, and sometimes the document corpus of the victim retriever as well. However, in practical scenarios, it's nearly impossible for the attacker to have access to the hidden documents corpus, which is held by the retrieval system behind, or to perturb the document immediately after receiving the explicit user's query at test-time. These limitations in practicability largely reduce existing works' impact on understanding the vulnerability of LLMR, and further drive us to raise the question: "*How vulnerable are LLM retrievers in the face of black-box adversarial attacks in practical scenarios*?"

To answer this question and address limitations of existing studies, we propose a complete *black-box* attack method, which learns transferable tokens on a surrogate LLM with only knowledge of victim documents, and particularly focuses on absenting victim information sources. To be specific, (1) We first establish a theoretical framework based on empirical observations, and analyze the three factors for achieving black-box transferable attack: derivation of victim document embedding, precision of surrogate LLM and precision of the target LLM; (2) To yield sufficient topic derivation, our attack samples potential user queries utilizing casual LLMs, and performs an adversarial learning between the sampled query and the victim documents on the surrogate model. (3) To further strengthen optimality, we adapts two technical tricks: imbalance on query population and word-embedding as surrogates. Compared with existing attacks, our method have three main advantages: **(1) Practical attacker capacity settings**, which simulate a potential attacker that can only read and slightly edit the knowledge documents as a normal user (e.g., Wikipedia contributor, reddit thread commenter) without any knowledge of retriever models, builders or users; **(2)Theoretical insights on attack transferability**, which is methodological deducted based on our established framework of *topic embedding clusters* of LLMs and formulated into necessary conditions for achieving transferable attacks on surrogate models; **(3) Zero-shot transferable attack over different LLM retrievers**, which doesn't require high interactions with a specific victim model like other black-box attacks, and once learnt, could be destructive against various LLM retrievers globally. In other words, once a document is polluted by our injected tokens, all retrievers would be possibly influenced to retrieve it at the same time.

This paper's contributions are summarized as follows:

1. We first investigate the vulnerability of LLMR in the face of **query-agnostic black-box settings**, where the attacker has no access to the victim model, the docu-

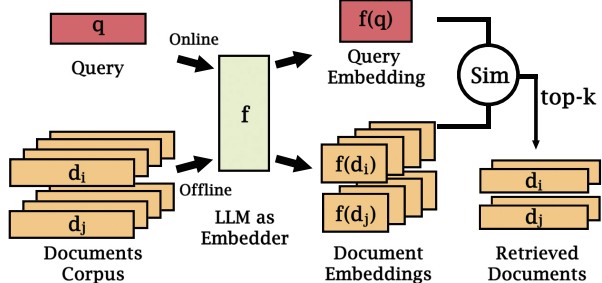

*Figure 2.* Illustration of LLM-based Retrieval. Documents in the corpus are firstly embedded in the last-hidden embeddings and stored. When a user query comes and get embedded, it matches relevant documents in embedding similarity in high efficiency.

ment corpus, or user queries in any kind of format.

2. We establish a **theoretical framework** on LLM retrievers with empirical verification, and propose an adversarial learning method between victim documents and sampled query candidates on a surrogate model with **zero-shot transferability**.

3. Our method is validated on benchmark datasets to be **globally effective across various** victim LLM retrievers with the **same** injected tokens for a specific document, and outperforms other existing attack baselines.

## 2. Preliminaries and Problem Formulation

**LLM as Retrievers**   Nowadays advancing large language models are mostly built on transformer architectures. For simplification of notation, in this paper we skip the setting of tokenizers, and represent all context datas (document or query) as a series of tokens. When a context $X \in \mathbb{R}^L$ is required for retrieval embedding on LLM $f$, it first forward into a word-embedding layer $f_0 : \mathbb{R} \to \mathbb{R}^{m_0}$:

$$\mathbf{X}_0 = f_0(X)$$

The processed $\mathbf{X}_0 \in \mathbb{R}^{L \times m_0}$ is called word-embeddings (later used in Section 3.3). $\mathbf{X}_0$ afterwards is forward into $T$ transformer blocks $f_{[T]} = \{f_1, \ldots, f_T\}$:

$$\mathbf{X}_i = f_i(\mathbf{X}_{i-1}), \forall i \in [1, T]$$

$\mathbf{X}_{[T]} \in \mathbb{R}^{L \times m}$ are described as the hidden in each block. When applying retrieval tasks, the last hidden $\mathbf{X}_T$ is adopted for the retrieval embedding through a last-hidden-pool function:

$$f(X) = \text{hidden-pool}(\mathbf{X}_T)$$

Depending on different settings, the pool function is usually a weighted sum in final attention or simply the last token hidden $\mathbf{X}_{T,L}$. When evaluate the relevance of two context $X_i$ and $X_j$, a statistic similarity function sim (usually cosine-similarity) is applied to $f(X_i)$ and $f(X_j)$. This statistic evaluation allows the retrieval system to preprocess

the embedding of the documents (usually in large amount), and only forward LLM once for the query when a user inquiries. This ensures the efficiency of the LLMR system when holding millions of documents prepared for retrieval.

**Token Injection Attack**   Token injection attack is one of the most popular approaches in LLM adversarial attack. For a given context $d$, the attacker injects a few attack tokens at the end of $d$ to achieve the LLM's performance decrease or unsafe behaviors like jailbreaks. Here we mainly introduce Greedy Coordinate Gradient (GCG), a white-box optimizable injection attack, for our information retrieval attack purpose. Specifically, the document $d$ is firstly injected with $m$ initial tokens (such as multiple $*$) as the initial $d'$:

$$d' = d \oplus t_{[m]}, \{t_i = \text{"*"}\}$$

Then the attacker loops greedy learning steps to optimize the injected tokens regarding a loss $\mathcal{L}$ depends on a specific task performed on $d'$. At each greedy step, $\mathcal{L}(g(d'))$ is backward to obtain the each token's gradients at the word embedding layer, which correspond to a specific gradient for every potential word $w$ (or token) in every $t_i$'s position:

$$\partial w^i = \frac{\partial \mathcal{L}(g(d'))}{\partial t_i}|_{t_i=w}, \forall w \in \mathcal{W}$$

$\mathcal{W}$ defines the whole vocabulary set that the victim LLM supports. Then the attacker pick up $k$ words $w^i_{[k]}$ that occupy top-$k$ greatest gradients $\partial w^i$ for every injected token $t_i$, and randomly choose one to replace the original one. Such a replacement constructs a greedily optimized sample $\{w_i\}$. The attacker makes several such samples, and eventually picks the one with lowest loss as the new $d'$.

$$t'_i \leftarrow w_i \in \arg\min_{\{w_i\}} \mathcal{L}(g(d \oplus \{w_i\}))$$

Note although GCG requires gradient backpropagation, the optimization is still a heuristic search in discrete forms.

**Problem Formulation**   Given a LLM model serving as a embedding function $f$, for a document $d$ belongs to a document corpus $D$, and a query document, the attacker manipulates the document to a perturbed version $d'$, which could be hard for the model $f$ to pick as top-$k$ from $D' = D \cup \{d'\} \setminus \{d\}$ in a embedding similarity order. We emphasize that in a **complete black-box** scenario, the capacity of attacker could be relatively limited:

1. The query $q$ is unknown to the attacker, since it's performed as a post-attack action by unknown victim user;

2. The LLM retriever model $f$ is a complete black-box, which means the attacker has no knowledge of $f$ including architecture and parameters, nor access to query it;

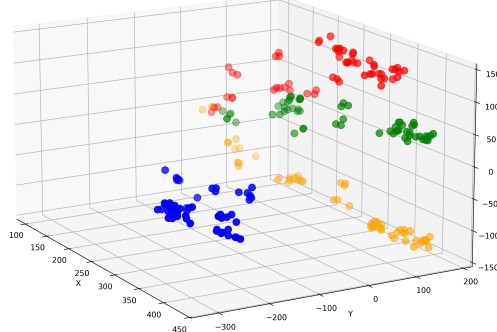

*Figure 3.* We sample 40 knowledge contexts on each of four roughly-defined topics and visualize their LLMR embeddings by Principal Component Analysis (PCA) reduction. Embeddings of contexts within the same topic (**R**:science, **G**:politic, **Y**:movie, **B**:architecture) tend to cluster together in the embedding space.

3. The document corpus $D$ is hidden to the attacker. The candidate documents are previously collected and hold by the retrieval system in an unknown manner;

4. The injected tokens applied to $d$ is limited in amount, which could be expressed as a distance constraint of $d$ and $d'$ with the function dis[1] and threshold $\delta$.

With the above settings, we formulate our attack problem as followed:

$$\arg\min_{d'} \mathbb{I}(d' \in \text{top}_k(\text{sim}(f(q), f(D'))))$$
$$s.t., D' = D \cup \{d'\} \setminus \{d\}, \text{dis}(d, d') \leq \delta \quad (1)$$

## 3. Query-Document Adversarial Learning

The main difficulty we faced in the stated problem is the black-box setting of victim LLM retriever, which gives neither parameters nor abundant query access. Therefore, we look for ways to learn our injection attack on on a surrogate LLM retriever. However, due to their high quantities in parameters, it is hard to build transferability between LLM models without enough insights. This hard-point draws us to do prior experiments to study the pattern of LLM embeddings, where we find a interesting yet rational phenomenon called **topic cluster**: the LLM-generated embeddings of context datas in the same topics tend to cluster, as shown in Figure 3. Based on this observation, (1) we first raise a theoretical definition called "$\epsilon - p_\epsilon$-Precise" retriever to quantify how LLM assembles the embeddings in same topics for the retrieval task, and verify this definition by simulation experiments. (2) Then utilizing this defined property, we are able to make methodology deduction on the transferability of the surrogate attack, and equalize it into a min-max problem. (3) Aiming to solve the min-max problem, we therefore design our Query-Document Adversarial Learning by optimizing both victim documents and sampled queries in an adversarial learning way. (4) Finally, we empirically find that by adopting the word-embedding for attack instead

---

[1] $\text{dis}(A, B) = |B| - |A|$ if $A \sqsubseteq B$; $\infty$ otherwise.

| Topic Name | | $p_0$ | $p_{0.1}$ | $p_{0.2}$ | $p_{0.3}$ | In Sim. | Out Sim. |
|---|---|---|---|---|---|---|---|
| Engin. | Chemical | 100% | 99.6% | 91.7% | 87.8% | [0.67,0.90] | [-0.07,0.58] |
| | Computer | 92.6% | 87.0% | 87.0% | 66.5% | [0.49,0.87] | [-0.05,0.63] |
| | Electric | 92.6% | 89.6% | 83.0% | 47.8% | [0.50,0.89] | [-0.05,0.63] |
| | Biology | 99.1% | 93.9% | 88.3% | 86.1% | [0.55,0.85] | [-0.09,0.56] |
| Movies | Action | 89.1% | 77.0% | 29.1% | 0.4% | [0.37,0.87] | [-0.06,0.60] |
| | Horror | 100% | 94.3% | 90.9% | 82.2% | [0.59,0.88] | [-0.04,0.59] |
| | Romantic | 88.3% | 87.4% | 79.6% | 28.7% | [0.44,0.88] | [-0.05,0.63] |
| | Comedy | 87.0% | 72.6% | 33.9% | 0.0% | [0.32,0.82] | [-0.07,0.63] |
| Achite. | Baroque | 99.6% | 94.3% | 91.3% | 87.4% | [0.70,0.94] | [-0.03,0.72] |
| | Gothic | 100% | 96.1% | 91.3% | 87.0% | [0.71,0.91] | [-0.05,0.71] |
| | Renaissance | 100% | 90.9% | 87.0% | 87.0% | [0.64,0.89] | [-0.02,0.72] |
| | Modern | 100% | 100% | 87.8% | 86.5% | [0.65,0.88] | [-0.03,0.54] |
| Geogra. | American | 84.8% | 78.7% | 50.4% | 5.2% | [0.40,0.86] | [-0.07,0.63] |
| | European | 92.2% | 87.0% | 86.5% | 78.3% | [0.57,0.89] | [-0.04,0.69] |
| | Asian | 89.1% | 87.0% | 87.0% | 80.0% | [0.54,0.85] | [-0.06,0.68] |
| | African | 91.3% | 87.4% | 87.0% | 83.0% | [0.58,0.85] | [-0.09,0.69] |

*Table 1.* Context topics' $p_\epsilon$ on Qwen1.5-7B-Ins. retriever. "In Sim" describes the minimum and maximum of similarity between every pair within the topic corpus ($10\times10$), and "Out Sim" for every pair of one within the topic and one in other 23 topics ($10\times230$). Full results on 24 topics and 3 LLMs are reported in Appendix E.

of last-hidden on LLM, the transferable attack works more effectively, which helps to strengthen our method.

### 3.1. Definition of $\epsilon$-$p_\epsilon$-Precise Retriever

Under the complete black-box setting, we don't know the specific inner properties of $f$, so we instead restrict its behavior as a well-performed retriever by assuming it always generates similar embeddings for contexts in the same topic while dissimilar ones for others:

**Definition 3.1** ($\epsilon$-$p_\epsilon$-Precise Retriever)**.** Given an global text set $\mathbb{X}$, a subset $\mathcal{X} \subseteq \mathbb{X}$ which contain contexts $\{X_1, X_2, ...\}$ within a same topic distribution, not limiting to query or document, and an complementary context set $\mathcal{X}' = \mathbb{X} - \mathcal{X}$ which consists of remaining contexts not in same topics with $\mathcal{X}$, we define a retriever to be $\epsilon$-$p_\epsilon$-Precise over $\mathcal{X}$ if there are at least $p_\epsilon$ fraction of $\mathcal{X}'$ (note as $\mathcal{X}'_\epsilon$) satisfying:

$$\text{sim}(f(X_i), f(X_j)) \geq \text{sim}(f(X_k), f(X')) + \epsilon, \\ \forall X_{i,j,k} \in \mathcal{X}, X' \in \mathcal{X}_\epsilon \quad (2)$$

The number $\epsilon \in [0, 1]$ describes a similarity gap between documents within and without the topics in probability of $p_\epsilon$. To verify the rationality of this statement, we conduct an experiment: we utilize a Casual LLM to generate 240 knowledge documents in 24 different topics (10 for each), and collect their embeddings on a LLM retriever. Then we test each topic's $p_0$, $p_{0.1}$, $p_{0.2}$ and $p_{0.3}$ against other 23 topics as Definition 3.1 described. Part of the result is reported in Table 1, where most of the topics exhibit in a tight embedding cluster with considerable $p_\epsilon$, while the full result is reported in Appendix E. This result effectively supports our assumption to be truthful. Under this definition, we assume the victim model $f$ is $\epsilon_f$-$p_\epsilon^f$-Precise over the topic $\mathcal{X}$ containing $d$ and $q$, and we have access to a surrogate model $g$

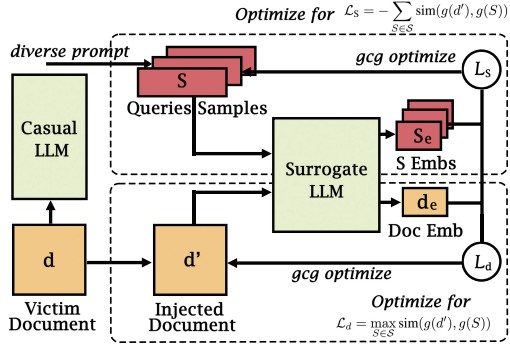

*Figure 4.* The DQ-A learning pipeline of our attack method. Query samples are first generated by a third party casual LLM. Then in every learning step, injected document tokens are first optimized away from queries, and all queries tokens are optimized towards the document. Both surrogate and Casual LLMs require no learning.

that is $\epsilon_g$-$p_\epsilon^g$-Precise over $\mathcal{X}$. With these assumptions, we then methodologically solve our attack problem.

### 3.2. Methodology for Adversarial Learning

First noticing the $D$ is unknown to us, and we also have no access to other document in $D$. Therefore, we transfer the attack target in Eq.1 to minimize the similarity between $d'$ and $q$:

$$\arg \min_{d'} \text{sim}(f(d'), f(q)) \quad (3)$$

Since we don't know the specific ground truth $q$ that the victim user uses, we instead surrogate Eq.3 by an upper bound:

$$\max_{X \in \mathcal{X}} \text{sim}(f(d'), f(X)) \geq \text{sim}(f(d'), f(q)) \quad (4)$$

Therefore, we seek to optimize $\max_{X \in \mathcal{X}} \text{sim}(f(d'), f(X))$ instead. Despite serving as an approximate bound of target query, we further analysis how this max term benefits the transferable of attack when we conduct on the surrogate $g$.

**Theorem 3.2** (Transferability of Attack on Victims)**.** *Assume $f$ and $g$ serve as two independent retrieval functions. After adversarial optimization, if $\max_{x \in \mathcal{X}} \text{sim}(g(d'), g(X))$ is optimized to no greater than $\min_{X_i, X_j \in \mathcal{X}} \text{sim}(g(X_i), g(X_j)) - \epsilon_g$ with a positive gap $\epsilon_g$, then $d'$ could be identified as $d' \in \mathcal{X}'$, which gives a bounded decrease on embeddings similarity of $f$ in probability of $p_\epsilon^g$. This transferability is formalized as:*

$$\underbrace{\max_{x \in \mathcal{X}} sim(g(d'), g(X))}_{\text{Optimization Objective}} \leq \underbrace{\min_{X_i, X_j \in \mathcal{X}} sim(g(X_i), g(X_j)) - \epsilon_g}_{\text{Lower Bound for Embeddings Within q's Topic}}$$

$$\overset{p_\epsilon^f}{\Longrightarrow} \underbrace{sim(f(d'), f(q))}_{\text{Transferred Attack Effect}} \leq \underbrace{\min_{X_i, X_j \in \mathcal{X}} sim(g(X_i), g(X_j)) - \epsilon_f}_{\text{Upper Bound for Embeddings Outside q's Topic}}$$

$$(5)$$

*Proof.* See in Appendix A.1. □

The above theorem indicates that as $f$ is better as the retrieval embedder, i.e., a larger $\epsilon_f$ with high $p_\epsilon^f$ exists, the transferred attack is more likely to be effective; and as $g$ is better as the embedder, i.e., a nonzero $\epsilon_f$ with high $p_\epsilon^f$ exists, the transferred attack is easier to achieve.

This maximum objective term aligns with the upper bound in Eq.4. Therefore we set it as our attack's optimization objective to replace Eq.3:

$$\arg\min_{d'}\max_{X\in\mathcal{X}}\mathrm{sim}(g(d'),g(X)) \qquad (6)$$

In practice, we cannot get all possible contexts within the topic $\mathcal{X}$ to calculate the maximum. Therefore, recalling the min-max problem solution raised in Generative Adversarial Nets (GAN) (Goodfellow et al., 2014), we instead approximate Eq.6 into two recursive equations:

$$\begin{aligned} X &= \arg\max_{X\in\mathcal{X}}\mathrm{sim}(g(d'),g(X)) \\ d' &= \arg\min_{d'}\mathrm{sim}(g(d'),g(X)) \end{aligned} \qquad (7)$$

To solve this pair of equations, we could first sample an initial query $X$ that is relevant to $d$ through LLM (optional to be $g$) generation, and adopt the classic adversarial learning to optimize the document and the query with their losses opposite on the similarity. However, we need to note that Eq.7 is not equivalent to Eq.6, but instead a necessary (or "weaker") condition to Eq.6. Intuitively, it just describes a first-order stationary point, and only when the $g$ function is convex on $\mathcal{X}$, it equals to the global min-max point of Eq.6, which is indeed hardly to happen in LLMR cases.

To better adapt our method in such nonconvex cases, we take reference to Two-Time-Scale GDA (Lin et al., 2020), which proves that it is helpful to set inner minimization steps larger than the outer maximization steps. This claim suggests us to apply different learning rate $\eta_d$ and $\eta_X$ as step sizes to the $d'$ and $X$, specifically with $\eta_X > \eta_d$. However in the GCG algorithm which is adopted to optimize $d$ and $X$, there exists two problems: (1) Each step of GCG learning is a heuristic search and no actual learning rate could be applied, which we have stated in Section 2; (2) The optimizable length of $X$ should be pre-set to achieve efficient GCG learning, however, would naturally limit the learning effect of $X$ and make its embedding hard to deviate much in the designed learning steps. Therefore, we choose to turn this step-scale difference into a population difference: instead of learning one sample $X$, we make $N$ diverse samplings:

$$\mathcal{S} = \{S_i = h(d \oplus p_i), i \in [1, N]\} \qquad (8)$$

where $p_{[N]}$ are crafted prompts to diversely generate related queries (described in Appendix), and $h$ is a third generative LLM. By optimizing each sampled query in $\mathcal{S}$ in fix steps and optimizing $X$ with the maximum of query's loss, our

Document-Queries Adversarial (DQ-A) learning turns Eq.7 into the adversarial losses:

$$\begin{aligned} \mathcal{L}_d &= \max_{S\in\mathcal{S}}\mathrm{sim}(g(d'),g(S)) \\ \mathcal{L}_S &= -\sum_{S\in\mathcal{S}}\mathrm{sim}(g(d'),g(S)) \end{aligned} \qquad (9)$$

In GCG learning, we optimize every token in the sampled queries for the loss $\mathcal{L}_S$, considering they don't need to be in practice use. While for the victim document, to simulate a normal user's edition, we follow the original GCG injection process that only optimize fixed injected tokens on $\mathcal{L}_d$ throughout the adversarial learning. Intuitively, this difference in tokens amount may also help for simulating $\eta_X > \eta_d$. Afterwards, the learnt adversarial tokens of $d'$ are the final injection attack tokens we select.

### 3.3. Surrogate Hidden with Word-Embedding

During the experiments we also empirically find an interesting phenomenon: if we use word-embedding (the first hidden by word-embedding layer) of the input document and the sampled queries to perform the adversarial learning as surrogate of the last hidden, it will offer an even more effective and transferable attack on the target black-box model $g$'s performance by retrieving with $g's$ last hidden. We analyze this may due to the abundant parameters of the LLM, and when projected back to the input context, the distance optimized in the latter layer is less influential than the distance optimized in the shallow layer, which decrease the former attack's effectiveness and transferability.

Therefore, we use the word-embedding of surrogate model $f$ instead of its last layer hidden to execute the steps in Section 3.2:

$$\begin{aligned} \tilde{g}(X) &= \mathbf{X}_0^g \in \mathbb{R}^{L\times m_0^g} \\ \tilde{\mathrm{sim}}(\tilde{g}(X),\tilde{g}(X')) &= \frac{\sum\mathrm{sim}(\mathbf{X}_{0,i}^g,\mathbf{X}_{0,j}'^g)}{LL'} \end{aligned} \qquad (10)$$

$\tilde{g}$ and $\tilde{\mathrm{sim}}$ stand for the word-embedding-surrogate version of embedding function and similarity matching metric. In the ablation study we will also compare the performance of DQ-A on word-embedding and on last-hidden.

## 4. Experiments

### 4.1. Experiment Setting

In our experiments, we use four datasets: "economics", "psychology", "biology" and "robotics", from a popular and challenging benchmark BRIGHT (Hongjin et al.) to validate our attack's influence on retrievers based on popular LLM models, including:

1. Classic opensource base models suggested by BRIGHT: Qwen-1.5-7B-Instruct, SFR-Embedding-Mistral and E5-Mistral-7B-Instruct;

2. High-download embedding LLM on huggingface: Embedding-Gemma-300M, Jina-Embeddings-v3, Granite-Embedding-r2 and Qwen3-Embedding-0.6B.

In ablation study, we test on three best-performance models: jinaai, gemma and Qwen1.5. We use the Qwen-1.5-4B-Casual both as our base surrogate model for learning attacks and casual model for generation. We adopt the word-embedding surrogate (Sec. 3.3) as default, and later analysis its impact in ablation study. For the baselines, we choose three typical attack methods: (1) Greedy Coordinate Gradient (GCG) (Kumar & Lakkaraju, 2024) on the same surrogate model to transfer attack, (2) PRADA (Wu et al., 2023) on the same surrogate model, and (3) Poisoned-RAG (Zou et al., 2025), who is prompted (described in Appendix) to generate tokens that diminish the document. Due to designed nature, we provide the ground-truth query for the three baselines. Additionally, we provide the victim model's query access and 10% of the queries/documents dataset for PRADA to do alignment training, which is achieved in LoRa finetuning process. Note while additional information creates a more beneficial setting for these baselines, the superiority of our method will be validated by following experiments even in face of such imbalanced comparison. We adopt 4 metrics for evaluation: NDCG@25, NDCG@50, Recall@25 and Recall@50. Recall measures if the ground-truth document is retrieved into the top 25/50, while NDCG suggests a rank-relevant metric. We apply both baselines our attacks on all ground-truth documents and evaluated how queries' retrieval performance drops compared with original results (lower is better ↓). We set the injected token amounts to 10 in our general experiment. For results not lower than 0.5%, we mark them in gray, indicating insufficient attacks.[2]

## 4.2. Experiments Results

**General Performance** The performance of LLM retrievers on original documents and attacked documents in the default setting is shown in Table 2. In general, our attack achieves the highest performance decrease compared with other attack methods. In 6 of 7 retrievers, our black-box attack can lead to an obvious decrease in all four averaged metrics, especially around a 2%-10% drop in Recall@25 and a 1%-7% drop in Recall@50. In dataset Robotics, our attack can achieve even an 8% performance drop for Qwen1.5 and 6% for Jinaai in Recall@50, which reduces these best performing retrievers' around 30% and 20% drop in fraction of their original performance. Only on Qwen3-Emb-0.6B, both our attack and other baselines find it hard to achieve a significant attack effect. We analyze that Qwen3-Emb is potentially post-trained against document injection attack for robustness, while this may also contribute to its lower performance compared with other small-scale LLMs like Jinaai

[2]Our implementation code is available at https://github.com/JetRichardLee/DQA-Learning.

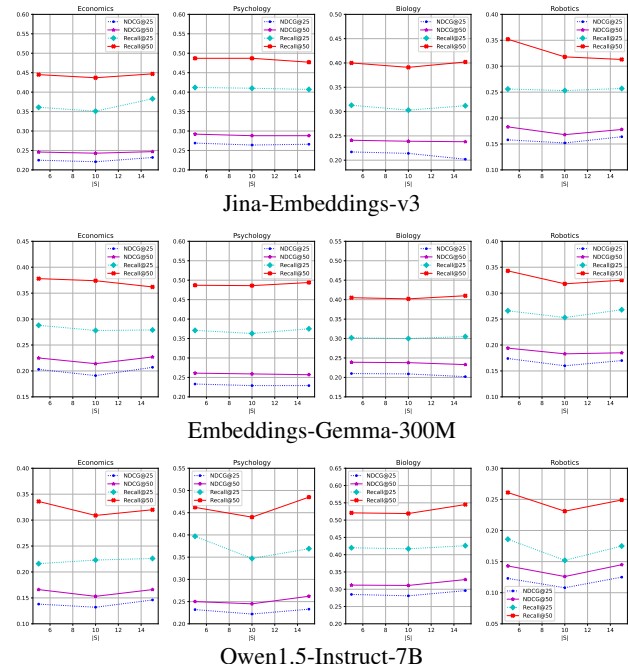

Jina-Embeddings-v3

Embeddings-Gemma-300M

Qwen1.5-Instruct-7B

*Figure 5.* Impact of Different $|\mathcal{S}|$.

and Emb-Gemma. This observed trade-off between robustness and precision further conforms with our Lemma 3.2 in Section 3.2. In conclusion, considering our strict limitation in injected tokens (suffix only) and no post-picking or crafting in attack samples, these decreasing effects constitute a strong evidence that LLMR could be vulnerable to adversarial attack even in complete black-box scenarios. We also notice that while Posioned-RAG behaves mostly bad, its oppositely helps to boosts the victim document even prompted to diminish it. One potential reason is that LLM generation task is mostly based on relevance, which may generate related tokens even required not to do so.

**Ablation Study of DQ-A Learning** To validate and study the effectiveness of our proposed adversarial learning framework and the word-embedding-surrogate strategy, we further compare our method with two ablation variates: DQ-A-H which doesn't apply word-embedding-surrogate strategy, and S-GCG which further doesn't apply adversarial learning but only GCG with sampled initial queries and summed similarities as the document loss. The test results are exhibited in Table 3, where we can find that DQ-A-H's attack generally decreases more performance than S-GCG, which validates the effectiveness of our adversarial learning setting. We also notice in a few columns DQ-A-H behaves a little worse, and analyze this may because the learned queries are not properly restricted to the original topic. Therefore while their embeddings chase to the polluted document and no longer lay in original topic cluster, their roles as topic-rejecters become less effective. Comparing the DQ-A and DQ-A-H, we find both baselines have some better attack

| Victim Models | Attacks | Economic | | | | Psychology | | | | Biology | | | | Robotics | | | | Average | | | |
| | | NDCG | | Recall | | NDCG | | Recall | | NDCG | | Recall | | NDCG | | Recall | | NDCG | | Recall | |
| | | @25 | @50 | @25 | @50 | @25 | @50 | @25 | @50 | @25 | @50 | @25 | @50 | @25 | @50 | @25 | @50 | @25 | @50 | @25 | @50 |
|---|---|---|---|---|---|---|---|---|---|---|---|---|---|---|---|---|---|---|---|---|---|
| QWen1.5-7B-Ins | Original | 0.212 | 0.237 | 0.314 | 0.409 | 0.272 | 0.292 | 0.405 | 0.493 | 0.336 | 0.359 | 0.469 | 0.553 | 0.158 | 0.177 | 0.245 | 0.311 | 0.245 | 0.266 | 0.358 | 0.442 |
| | GCG* | 0.184 | 0.209 | 0.291 | 0.393 | 0.259 | 0.281 | 0.398 | 0.493 | 0.330 | 0.355 | 0.468 | 0.552 | 0.138 | 0.152 | 0.212 | 0.259 | 0.228 | 0.249 | 0.342 | 0.424 |
| | PRADA* | 0.189 | 0.215 | 0.261 | 0.368 | 0.263 | 0.291 | 0.390 | 0.501 | 0.331 | 0.357 | 0.469 | 0.564 | 0.134 | 0.155 | 0.194 | 0.268 | 0.229 | 0.255 | 0.329 | 0.425 |
| | Poi.-RAG* | 0.233 | 0.260 | 0.340 | 0.454 | 0.308 | 0.327 | 0.446 | 0.527 | 0.357 | 0.385 | 0.493 | 0.596 | 0.229 | 0.251 | 0.309 | 0.399 | 0.282 | 0.306 | 0.397 | 0.494 |
| | DQ-A(Ours) | **0.132** | **0.153** | **0.223** | **0.309** | **0.222** | **0.245** | **0.347** | **0.440** | **0.281** | **0.311** | **0.417** | **0.519** | **0.108** | **0.126** | **0.152** | **0.231** | **0.186** | **0.209** | **0.285** | **0.375** |
| SF-Mis.-7B | Original | 0.203 | 0.224 | 0.311 | 0.395 | 0.235 | 0.261 | 0.418 | 0.519 | 0.238 | 0.270 | 0.348 | 0.476 | 0.187 | 0.198 | 0.289 | 0.321 | 0.216 | 0.238 | 0.342 | 0.428 |
| | GCG* | 0.199 | 0.220 | 0.303 | 0.396 | 0.239 | 0.266 | 0.441 | 0.542 | 0.227 | 0.258 | 0.338 | 0.453 | 0.159 | 0.177 | **0.245** | 0.312 | 0.206 | 0.230 | 0.332 | 0.426 |
| | PRADA* | 0.199 | 0.219 | 0.315 | 0.403 | 0.235 | 0.260 | 0.431 | 0.526 | 0.217 | 0.256 | 0.317 | 0.459 | 0.161 | 0.176 | 0.249 | **0.309** | 0.203 | 0.228 | 0.328 | 0.424 |
| | Poi.-RAG* | 0.242 | 0.263 | 0.346 | 0.434 | 0.269 | 0.289 | 0.455 | 0.529 | 0.249 | 0.280 | 0.365 | 0.467 | 0.272 | 0.290 | 0.360 | 0.418 | 0.258 | 0.281 | 0.382 | 0.462 |
| | DQ-A(Ours) | **0.188** | **0.210** | **0.297** | 0.392 | **0.213** | **0.243** | **0.384** | **0.495** | **0.205** | **0.234** | **0.310** | **0.417** | **0.149** | **0.171** | 0.262 | 0.314 | **0.189** | **0.215** | **0.313** | **0.405** |
| e5-Mis.-7B | Original | 0.182 | 0.198 | 0.294 | 0.365 | 0.199 | 0.229 | 0.346 | 0.457 | 0.225 | 0.259 | 0.326 | 0.451 | 0.183 | 0.197 | 0.269 | 0.311 | 0.197 | 0.221 | 0.309 | 0.396 |
| | GCG* | 0.158 | 0.183 | 0.252 | 0.355 | 0.190 | 0.224 | 0.324 | 0.462 | 0.212 | 0.242 | 0.311 | 0.417 | 0.137 | 0.159 | **0.216** | 0.301 | 0.174 | 0.202 | 0.276 | 0.384 |
| | PRADA* | 0.166 | 0.186 | 0.266 | 0.348 | 0.184 | 0.213 | 0.323 | 0.431 | 0.203 | 0.231 | 0.300 | 0.401 | 0.142 | 0.166 | 0.217 | 0.302 | 0.174 | 0.199 | 0.277 | 0.371 |
| | Poi.-RAG* | 0.197 | 0.220 | 0.305 | 0.404 | 0.225 | 0.246 | 0.377 | 0.465 | 0.229 | 0.266 | 0.324 | 0.461 | 0.272 | 0.288 | 0.366 | 0.419 | 0.231 | 0.255 | 0.343 | 0.437 |
| | DQ-A(Ours) | **0.145** | **0.173** | **0.237** | **0.349** | **0.161** | **0.194** | **0.280** | **0.410** | **0.188** | **0.211** | **0.287** | **0.363** | **0.125** | **0.145** | 0.224 | **0.295** | **0.155** | **0.181** | **0.257** | **0.354** |
| Emb.-Gemma | Original | 0.225 | 0.245 | 0.307 | 0.395 | 0.252 | 0.285 | 0.395 | 0.523 | 0.223 | 0.254 | 0.313 | 0.420 | 0.178 | 0.197 | 0.262 | 0.336 | 0.220 | 0.245 | 0.319 | 0.419 |
| | GCG* | 0.203 | 0.222 | 0.293 | 0.375 | 0.234 | 0.267 | **0.360** | 0.488 | 0.213 | 0.244 | 0.304 | 0.414 | **0.156** | **0.180** | 0.240 | 0.330 | 0.202 | 0.228 | **0.299** | **0.402** |
| | PRADA* | 0.207 | 0.229 | 0.285 | 0.381 | 0.243 | 0.265 | 0.375 | 0.487 | 0.213 | **0.242** | 0.307 | 0.416 | 0.157 | 0.181 | **0.232** | **0.324** | 0.205 | 0.229 | 0.300 | **0.402** |
| | Poi.-RAG* | 0.240 | 0.272 | 0.308 | 0.441 | 0.278 | 0.312 | 0.456 | 0.561 | 0.231 | 0.262 | 0.330 | 0.440 | 0.217 | 0.237 | 0.322 | 0.393 | 0.242 | 0.271 | 0.354 | 0.459 |
| | DQ-A(Ours) | **0.191** | **0.214** | **0.278** | **0.374** | **0.229** | **0.259** | 0.363 | **0.486** | **0.209** | **0.238** | **0.300** | **0.402** | 0.160 | 0.183 | 0.251 | 0.334 | **0.198** | **0.224** | 0.298 | 0.399 |
| jina-Emb.-v3 | Original | 0.225 | 0.248 | 0.360 | 0.453 | 0.274 | 0.295 | 0.422 | 0.489 | 0.222 | 0.249 | 0.325 | 0.422 | 0.176 | 0.200 | 0.275 | 0.371 | 0.224 | 0.248 | 0.346 | 0.434 |
| | GCG* | 0.223 | 0.246 | 0.370 | 0.459 | 0.274 | 0.295 | 0.422 | 0.482 | 0.213 | 0.237 | 0.317 | 0.399 | 0.162 | 0.184 | 0.261 | 0.346 | 0.218 | 0.241 | 0.343 | 0.422 |
| | PRADA* | 0.225 | 0.245 | 0.373 | 0.450 | 0.273 | 0.299 | 0.413 | 0.497 | **0.211** | **0.236** | 0.316 | 0.404 | 0.169 | 0.184 | 0.268 | 0.329 | 0.220 | 0.241 | 0.343 | 0.420 |
| | Poi-RAG* | 0.241 | 0.264 | 0.383 | 0.473 | 0.294 | 0.320 | **0.406** | 0.503 | 0.214 | 0.245 | 0.317 | 0.427 | 0.235 | 0.261 | 0.328 | 0.425 | 0.246 | 0.273 | 0.356 | 0.457 |
| | DQ-A(Ours) | 0.221 | **0.243** | **0.351** | **0.437** | **0.264** | **0.288** | 0.410 | 0.487 | 0.214 | 0.239 | **0.303** | **0.391** | **0.152** | **0.168** | **0.253** | **0.318** | **0.213** | **0.235** | **0.329** | **0.408** |
| Granite-Emb.-R2 | Original | 0.187 | 0.210 | 0.241 | 0.342 | 0.182 | 0.203 | 0.312 | 0.396 | 0.158 | 0.181 | 0.242 | 0.325 | 0.128 | 0.146 | 0.200 | 0.272 | 0.164 | 0.185 | 0.249 | 0.334 |
| | GCG* | 0.174 | 0.198 | 0.233 | 0.335 | 0.182 | 0.203 | 0.325 | 0.398 | 0.189 | 0.220 | 0.298 | 0.402 | 0.114 | 0.132 | 0.203 | 0.270 | 0.165 | 0.188 | 0.265 | 0.351 |
| | PRADA* | 0.182 | 0.200 | 0.255 | 0.338 | 0.180 | 0.201 | 0.318 | 0.392 | 0.159 | 0.183 | 0.239 | 0.326 | **0.108** | **0.132** | **0.175** | 0.257 | 0.157 | 0.179 | 0.247 | 0.328 |
| | Poi.-RAG* | 0.196 | 0.221 | 0.256 | 0.364 | 0.188 | 0.214 | 0.316 | 0.409 | 0.159 | 0.186 | 0.236 | 0.330 | 0.162 | 0.184 | 0.234 | 0.321 | 0.176 | 0.201 | 0.261 | 0.356 |
| | DQ-A(Ours) | **0.157** | **0.179** | **0.231** | **0.326** | **0.177** | **0.197** | 0.312 | **0.386** | **0.152** | **0.179** | **0.229** | **0.326** | 0.111 | 0.130 | 0.183 | **0.254** | **0.149** | **0.171** | **0.239** | **0.323** |
| Qwen3-Emb.-0.6B | Original | 0.198 | 0.211 | 0.293 | 0.344 | 0.197 | 0.218 | 0.343 | 0.419 | 0.159 | 0.177 | 0.242 | 0.305 | 0.133 | 0.149 | 0.203 | 0.264 | 0.172 | 0.189 | 0.270 | 0.333 |
| | GCG* | 0.194 | 0.208 | **0.288** | 0.342 | 0.203 | 0.223 | 0.341 | **0.412** | 0.152 | 0.168 | **0.215** | 0.296 | 0.135 | 0.150 | 0.204 | 0.259 | 0.171 | 0.187 | **0.262** | **0.327** |
| | PRADA* | **0.192** | **0.205** | **0.288** | 0.350 | 0.203 | 0.224 | 0.340 | 0.413 | 0.149 | 0.169 | 0.219 | **0.292** | 0.136 | 0.148 | 0.216 | 0.258 | 0.170 | 0.187 | 0.266 | 0.328 |
| | Poi.-RAG* | 0.204 | 0.222 | 0.294 | 0.363 | 0.207 | 0.341 | 0.340 | 0.447 | 0.156 | 0.176 | 0.234 | 0.305 | 0.175 | 0.193 | 0.269 | 0.333 | 0.186 | 0.233 | 0.284 | 0.362 |
| | DQ-A(Ours) | **0.192** | 0.212 | **0.288** | 0.359 | 0.209 | 0.230 | 0.341 | 0.417 | **0.146** | **0.165** | 0.226 | **0.292** | **0.128** | **0.142** | 0.200 | **0.246** | 0.169 | 0.187 | 0.264 | 0.329 |

*Table 2.* Performance drops after attack(↓). The best results among baselines are marked in **bold**. To better visualize how challenging the task is we also mark results with decrease less than 0.005 in gray. The last four columns are the average results of the former four datasets for general comparison. "*" means this method requires the ground-truth query for the victim document.

| | Attacks | Economics | | Psychology | | Biology | | Robotics | |
| | | R@25 | R@50 | R@25 | R@50 | R@25 | R@50 | R@25 | R@50 |
|---|---|---|---|---|---|---|---|---|---|
| jina | S-GCG | 358 | 448 | 0.414 | 0.484 | 0.312 | 0.422 | 0.272 | 0.328 |
| | DQ-A-H. | 0.360 | 0.444 | 0.412 | 0.484 | 0.308 | 0.426 | 0.245 | 0.323 |
| | DQ-A | 0.351 | 0.437 | 0.410 | 0.487 | 0.303 | 0.391 | 0.253 | 0.318 |
| Gemma | S-GCG | 0.297 | 0.374 | 0.396 | 0.496 | 0.302 | 0.423 | 0.268 | 0.342 |
| | DQ-A-H. | 0.282 | 0.360 | 0.378 | 0.495 | 0.308 | 0.408 | 0.238 | 0.344 |
| | DQ-A | 0.278 | 0.374 | 0.363 | 0.486 | 0.300 | 0.402 | 0.251 | 0.334 |
| Qwen | S-GCG | 0.299 | 0.376 | 0.407 | 0.520 | 0.478 | 0.588 | 0.181 | 0.284 |
| | DQ-A(H) | 0.289 | 0.401 | 0.382 | 0.511 | 0.482 | 0.537 | 0.214 | 0.286 |
| | DQ-A | 0.223 | 0.309 | 0.347 | 0.440 | 0.417 | 0.519 | 0.152 | 0.231 |

*Table 3.* Performance for Ablation Baselines(↓).

results, while DQ-A slightly outperforms. This comparison indicates that while the word-embedding strategy has effects as Section 3.3 states, most of the effectiveness still comes from our adversarial learning design.

**Impact of Sampling Amount** To study the impact of the amount of sample queries in our DQ-A learning and validate the population effect we raised in Section 3.2, we also test the decrease effect of our method under different $|\mathcal{S}|$. As Figure 5 illustrates, when $|\mathcal{S}|$ increases from 5 to 10, attacked performances on all LLM retrievers and datasets decrease more in different scales. This observation accords with our motivation on increasing sample population to achieve more optimal solution in non-convex cases. However, when $|\mathcal{S}|$ increases from 10 to 15, some of attacks become less effective. From this phenomenon we infer that there is also a limit in $|\mathcal{S}|$'s population effect. When $|\mathcal{S}|$ is larger than the inherent optimal $\frac{\eta_X}{\eta_d}$'s require, attacks become less optimal.

**Impact of Injected Token Amount** We also study the impact of the injected token amount (constrained by the $\delta$ in Eq.1) on our attack and explore its effect when the attacker can afford more injection budgets. As Figure 6 illustrates, however, it's surprising to find that the attack effects only shift around slightly without significant variations. This result suggests that for the token injection attack on paragraph embedding, a high length of suffix injected tokens length may not have much effect on embedding shifts compared with a fitted length.

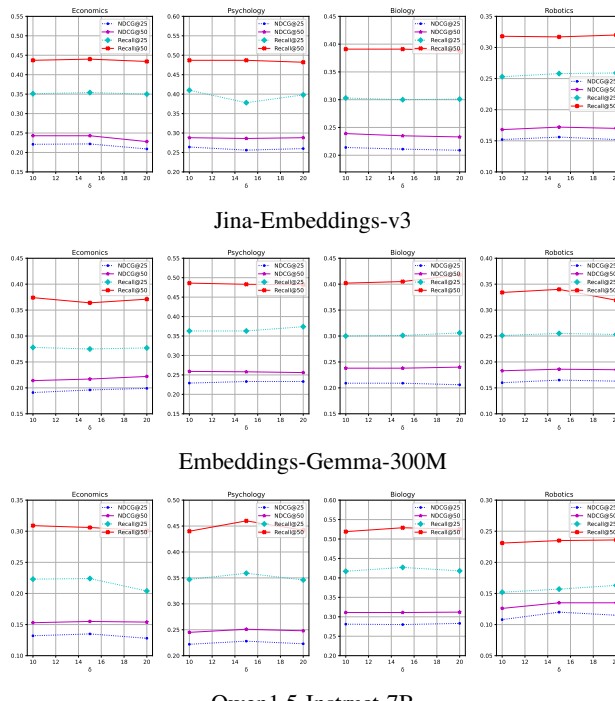

Jina-Embeddings-v3

Embeddings-Gemma-300M

Qwen1.5-Instruct-7B

*Figure 6.* Impact of Injected Token Length (↓).

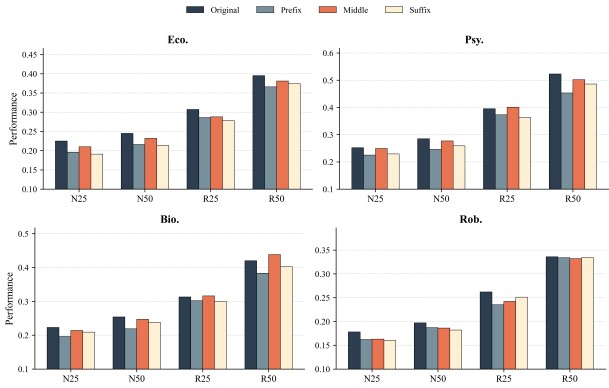

*Figure 7.* Impact of Injection Position(↓).

| Comb. | Attack | Eco. | | | Psy. | | | Bio. | | | Rob. | | |
|---|---|---|---|---|---|---|---|---|---|---|---|---|---|
| | | N25 | N50 | N100 | N25 | N50 | N100 | N25 | N50 | N100 | N25 | N50 | N100 |
| BGE- | Orig. | 0.139 | 0.160 | 0.165 | 0.175 | 0.190 | 0.209 | 0.132 | 0.156 | 0.172 | 0.128 | 0.130 | 0.143 |
| M3s | DQ-A | 0.131 | 0.151 | 0.160 | 0.166 | 0.179 | 0.197 | 0.121 | 0.145 | 0.164 | 0.109 | 0.122 | 0.135 |
| Gem.+ | Orig. | 0.212 | 0.230 | 0.263 | 0.236 | 0.264 | 0.276 | 0.249 | 0.280 | 0.323 | 0.175 | 0.188 | 0.211 |
| Qwen3 | DQ-A | 0.182 | 0.200 | 0.234 | 0.222 | 0.245 | 0.264 | 0.231 | 0.253 | 0.289 | 0.181 | 0.183 | 0.203 |

*Table 4.* Performance drops after attack on retriever+reranker combinations(↓).

**Impact of Attack Position**  While in most token injection attack settings the injected position is typically chosen as the end of the input context, this setting does not reflect all cases in practice, e.g., the web context may be dynamically updated and extended by other normal users. Therefore, we conduct the ablation study on the chosen injection positions on Gemma. Specifically, we inject the adversarial tokens at the (1)beginning, (2)middle point (at half length), and (3)end of victim documents, and compare their retrieval performance with the original, which is illustrated in Figure 7. We notice when our generated token is injected at the middle point, the attack performance is the worst, and much better at the beginning and at the end. This observation aligns with the common sense that the beginning context and the nearest part (ending) usually occupy higher attention than middle part in transformer-based models.

**Retriever with Reranker**  Despite the sole retrieval task, in some application scenarios the LLM retriever is be used with a reranker model together. Specifically, the retriever first picks a candidate pool from the whole document corpus, and the reranker then scores precisely within the pool and refine the ranking order. While this two-stages architecture achieves higher ranking performance compared with a sole retriever, we clarify it cannot defend against our attack: our attack focuses on the upper retrieval stage, aiming to filter the victim document out of the pool firsthand, and therefore however strong a reranker is applied afterwards, the document is already out of retrieval. To validate this statement, we additionally test two retriever-reranker combi-

nations. The retriever model first selects the top 25/50/100 documents, filtering out others, and then the reranker model scores retrieved documents and reports the NDCG25/50/100. The first set is the BGE-M3 retriever with BGE-reranker-v2-M3, and the second is Gemma-300M with Qwen3-reranker-0.6B. The performance results are illustrated in Table 4.

**Computational Cost and Complexity**  In Table 5 we provide the time complexity of our attack and its empirical time cost on average of test datasets. $V$ and $D_{dim}$ refers to the vocabulary size and the dimension size of the surrogate LLM, and $|d|$ refers to the victim document length in token amount. Since our attack aims to decrease the retrieved possibility of a specific victim document instead of creating batches of corruption documents, this attack computation is generally acceptable.

**Approximating Verification for Theorem 3.2**  As we claim the Theorem 3.2, it is necessary to verify its establishment in empirical practice. However, since the concept of *topic set* is defined as an infinite context set, it is impossible to verify a precise $p_\epsilon$ on the victim document's topic set. Therefore, we conduct an approximating verification by restricting a countable topic set, i.e., the set only contains the victim document and query. Since our definition of $\epsilon - p_\epsilon$ always holds for any context set, this approximating verification is rational and sufficient to support our theoretical claim. The full verification process, approximation deduction, and verification result are provided in Appendix A.2.

| Time Complexity Per Epoch | Average Time Per Document |
|---|---|
| $\mathcal{O}(|\mathcal{S}|(|d| + \delta)^2 D_{dim} + \delta V)$ | 23.065s($\delta = 10$, 20 epoches) |

*Table 5.* Computational Cost and Complexity of DQA.

# 5. Related Works

**Adversarial Attack on Retriever Systems** In information retrieval systems, adversarial attack methods generally have two categories: (1) Creating corrupted documents to the corpus and influence the retriever performance on normal documents (Liu et al., 2023a; Lin et al., 2024; Long et al., 2024; Zhong et al., 2023a); (2) Manipulating the victim document to decrease or increase its retrieval possibility (Wu et al., 2023; Liu et al., 2023b). With the advance of RAG, adversarial attack against such mechanism also arises attention, which mostly focuses on (3) Manipulating or creating a virus document to be highly likely retrieved and poison the generation process (Xiang et al., 2024; Liang et al., 2025; Zou et al., 2025), leading to LLM's unsafe behaviors. Recently some works on agent memory safety also propose methods on (4) Crafting or Manipulating memory entries with poisoning information to be likely retrieved by the LLM retrievers (Chen et al., 2024; Dong et al., 2025).

**Token Injection Attack on Large Language Model** Injection attacks against LLM is one of the most popular attack type on LLM robustness study. Technically these attack could be summarized into the three categories on crafting methods: (1) Human-crafted attack (Greshake et al., 2023; Shen et al., 2024; Schulhoff et al., 2023), which is mostly created by human, including amateurs from internet and LLM researchers. (2) Token-optimization-based attack (Jia et al.; Hayase et al., 2024; Zou et al., 2023; Geisler et al.), which designs specific losses on the LLM downstream task such like generation and retrieval, and trigger specific desired results by heuristically search the optimal combination of injected tokens' word choices. (3) LLM generation attack (Yu et al., 2023; Mehrotra et al., 2024; Chao et al., 2025), which requires a Casual LLM to generate adversarial tokens automatically. Such query-based attack is usually easy to achieve, while also found effective in many cases.

# 6. Discussion

*Q: Could this diminishing attack be extended to boost specific document?* The success of our proposed attack is mainly relying on "rejecting documents" from its original relevant groups. Therefore on the boosting task, (1) if a manipulated document is hoped to be retrieved with **an irrelevant query**, (e.g., recommending toy selling websites to queries on a cartoon, indicating a potential child user), our attack is likely to work with adversarial learning loss to be reversed; (2) however, if a manipulated document is hoped to be retrieved by **relevant queries with higher rankings**, our attack may be not promising since it aims for "out-of-group" relevance rather than "within" relevance. To further validate the statement (1), we also conduct a small test: we first randomly pick a document from the original economics dataset's corpus, and change all queries' ground truth to be

only the picked document. This setting firstly results in a nearly zero performance, with Recall@50 only 0.00227 and NDCG@50 only 0.00116. Then we reverse the losses of our attack, change the maximum term respect to sampled queries to minimum terms, and apply the reversed attack to the picked document. The Recall@50 is then raised to 0.00601 and NDCG@50 to 0.00251. Although this doubled effect is still limited in lack of further refining, it shows that it is possible to use our attack for enhancing the document retrieval, and we will improve and explore this reversed method further in the future work.

*Q: Is there defense method for our proposed attack?* Existing defense methods on retriever systems mainly focus on corpus poisoning attack (Hong et al., 2024; Wu et al., 2022), i.e., one or several adversarial documents are injected into the corpus, and get boosted into the retrieved list. This setting naturally leads these works to focus on detecting poisoned retrieved documents (Pathmanathan et al., 2025; Zhong et al., 2023b), or ensuring robust RAG generation (Xiang et al., 2024), however, leaving the documents not retrieved in the shadow. This post-retrieval characteristic makes these attacks naturally not applicable against our decreasing attack. Despite these specifically designed retrieval defense, there are also perplexity detecting methods (Alon & Kamfonas, 2023; Hu et al., 2023; Li et al., 2026) against LLM token injections, however, also validated to be insufficient for our attack in Appendix B.1. Moreover, in our experimental section there are victim models like Gemma and Jinaai-v3 which already apply preprocess techniques (e.g., truncation) inside, also failing to defend. In all, to our best knowledge, there is no defense method applicable for our attack so far. As we notice that in other domains there exist approaches utilizing majority voting on subcontent for item-level ranking defense (Li et al.; Li & Wang, 2025), we will explore adaption of these relevant studies to provide document-level defense in our future work.

# 7. Conclusion

In this paper we study the vulnerability of LLM-based retrieval by proposing a query-agnostic black-box attack, which requires no knowledge of victim model parameters, access nor the target victim query, and stays effective with zero-shot transferability against different retrievers at the same time. To achieve our attack, we first establish a theoretical framework to describe LLM retrievers' property and verify it empirically. Then based on this framework, we analyze the transferability of the surrogate model attack, and design an adversarial learning mechanism with a word-embedding-surrogate strategy. Our proposed attack is validated to be effective against most popular LLMR models on challenging benchmark datasets. Through this work, we further call for defense methods that not only focus on post-retrieval detections but also build robust retrieval itself.

## Impact Statement

This paper studies the robustness of LLM-based retrieval systems by identifying a previously underexplored failure mode under realistic black-box conditions. The primary goal of this work is to *advance the understanding of retrieval behavior* in modern LLM-based systems. The techniques analyzed in this paper are intended to characterize system vulnerabilities rather than to promote adversarial misuse. Similar retrieval failures may arise unintentionally in practice due to benign document edits, formatting changes, or content updates. By systematically studying such behaviors, this work aims to inform the development and evaluation of more robust retrieval systems.

This research does not involve human subjects, personal data, or user interaction logs. All experiments are conducted on public benchmark datasets using open-source models. We do not identify additional societal consequences beyond those commonly associated with research on system robustness and security in machine learning.

## Acknowledgments

This work was partially supported by the National Science Foundation under Award No. 2428039, No. 2346158, No. 2449280. We also acknowledge the use of computational resources provided by the Advanced Cyberinfrastructure Coordination Ecosystem (Boerner et al., 2023): Services & Support (ACCESS) program, supported by NSF grants #2138259, #2138286, #2138307, #2137603, and #2138296. Specifically, this work used the NCSA Delta GPU at the National Center for Supercomputing Applications (NCSA) through allocations CIS251004, CIS260196, and CIS250765. The work is also partially supported by Amazon Research Awards. Any opinions, findings, conclusions, or recommendations expressed in this material are those of the authors and do not necessarily reflect the views of the National Science Foundation and Amazon. This work was also supported in part by the Department of Defense under Cooperative Agreement Number W911NF-24-2-0133. The views and conclusions contained in this document are those of the authors and should not be interpreted as representing the official policies, either expressed or implied, of the Army Research Office or the U.S. Government. The U.S. Government is authorized to reproduce and distribute reprints for Government purposes notwithstanding any copyright notation herein.

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

# A. Proofs

## A.1. Proof for Theorem 3.2

On the surrogate model $g$, when the maximal similarity of the injected document $d'$ with the topic cluster set is lower than the minimal similarity within the topic cluster with a positive gap value $\epsilon_g$, we could infer the document is injected outside the set:

$$(\max_{X \in \mathcal{X}} \text{sim}(g(d'), g(X)) \leq \min_{X_i, X_j \in \mathcal{X}} \text{sim}(g(X_i), g(X_j)) - \epsilon_g) \wedge (\epsilon_g > 0) \implies d' \notin \mathcal{X}$$

Since the victim model $f$ is black-box model to us, we can only assume it as an independent embedding function with the surrogate model $g$. However, as the victim model $f$ follows the $\epsilon_f - p_\epsilon^f$ precision on this cluster set $\mathcal{X}$, which describes a probabilistic property for documents on non-relevant topic distribution, we could simply estimate outside $d'$ document a probabilistic bound by this property:

$$d' \notin \mathcal{X} \xRightarrow{p_\epsilon^f} \text{sim}(f(d'), f(q)) \leq \max_{X \in \mathcal{X}} \text{sim}(f(d'), f(X)) \leq \max_{X_i, X_j \in \mathcal{X}} \text{sim}(f(X_i), f(X_j)) - \epsilon_f$$

This two-step deduction proves our theorem.

## A.2. Approximation Deduction for Verification

In this section we provide a further empirical verification on the transferability lemma to show its rationality in practice.

**How we verify**   To be specific, for all tuples of query $q$, groundtruth document $d$ and attacked $d'$ in our main experiment results, we first pick up the tuples that satisfy:

$$\max(\text{sim}(g(d'), g(q)), g(d)) \leq \text{sim}(g(d), g(q))$$

and count how much fraction of these tuples satisfy the following inequality for a given $\epsilon_f$, which we choose as 0, 0.05 in this verification experiment:

$$\text{sim}(f(d'), f(q)) \leq \text{sim}(f(d), f(q)) - \epsilon_f$$

**Why could verify**   Under this evaluation, we treat $d$ and $q$ as an extremely small topic corpus $\mathcal{X} = \{q, d\}$. Noting such a weak corpus may not have a positive $\epsilon_g$ with $p_\epsilon^g = 1$, without loss of generalization, we simply give relaxation on the left hand of the transferability with removal of $\epsilon_g > 0$ and verify for:

$$\max_{X \in \mathcal{X}} \text{sim}(g(d'), g(X)) \leq \min_{X_i, X_j \in \mathcal{X}} \text{sim}(g(X_i), g(X_j)) = \text{sim}(g(q), g(d))$$

$$\xRightarrow{p_\epsilon^f} \text{sim}(f(d'), f(q)) \leq \min_{X_i, X_j \in \mathcal{X}} \text{sim}(g(X_i), g(X_j)) - \epsilon_f = \text{sim}(f(d), f(q)) - \epsilon_f$$

To verify the practical $p_\epsilon^f$ probability by expectation, we only need to count how much fraction of tuples satisfying the left hand could also satisfy the righthand, and that's exact what we do in the above statement. We emphasize that such relaxation on left hand dors not reduce the strongness of verification, instead, posing an approximation and potentially more hardness.

**Verification results**   The evaluation results of each dataset are illustrated in Table 6: Given the relatively small topic corpus constructed with low $\epsilon - p_\epsilon^f$ and hardness brought by left hand relaxation, our considerable verifed probability on $\epsilon_f = 0$ and non-zero probability on $\epsilon_f = 0.05$ could partially prove the rationality of our theoretician deduction steps and conclusion.

| Data. | Eco. | Psy. | Bio. | Rob. |
|---|---|---|---|---|
| $p_0^f$ | 80.0% | 69.7% | 72.0% | 58.9% |
| $p_{0.05}^f$ | 7.6% | 3.7% | 0.9% | 1.4% |

*Table 6.* Verification result on each dataset.

# B. Additional Experiments

## B.1. Perplexity Detection Defense

Detection based methods have been a popular approach for selecting malicious inputs from large distributional data, which are proved effective against both LLM token injection (Hu et al., 2023; Alon & Kamfonas, 2023) and general OOD cases (Li et al., 2025b;a) in other domains. In this study we conduct an evaluation on perplexity of our attacked documents, and compare with the original ones. Specifically, we compare with (1) corresponding original document perplexity, (2) the 100th largest and (3) the largest perplexity of the original corpus (each dataset 50K around docs). The results are illustrated in Table 7. We first notice that indeed our attack increases all victim documents perplexity. However, comparing with the

| Compared | Eco. | Psy. | Bio. | Rob. |
|----------|------|------|------|------|
| Original | 100% | 100% | 100% | 100% |
| 100th L. | 7.3% | 3.5% | 11.4% | 2.7% |
| Largest | 0.0% | 0.0% | 0.0% | 0.0% |

*Table 7.* Perplexity Comparison

whole original corpus, most of our attacked documents do not override the highest perplexity part (top 100), and none of them exceeds the highest one. This suggests our attacked documents are still in the original corpus' natural perplexity range without outstanding gap. Noting our attack is on a single document rather than creating a set of documents, it is hard to be detected: to detect the attacked document by perplexity, even if the detector blindly removes top 100 perplexity documents, ours is still likely to survive. Therefore, the perplexity detection is not effective for our attack. We also present a shortened attacked sample:

> Evidence[edit] Growth and aging[edit] There is a large body of evidence ... (7012 characters except space) ... through clearance and recycling of damaged proteins and organelles..basicConfig useContext.dispatchEvent ArgumentOutOfRangeException corres SNMP sede masturb-Javadoc lineno

This attacked example is from Bio. dataset. Compared with the original long context, our injected tokens are short and have no outstanding characters. This explains why the perplexity increase is not obviously high.

# C. Prompt Designs

## C.1. Prompts for Theoretical Verification

To verify the rationality of the $\epsilon$-$p_\epsilon$-Precise retriever definition, we utilize the following prompts to generate 240 knowledge documents in 24 different topics:

```
topics = ["Chinese Politics","Japanese Politics","American Politics","English Politics", "Comedy Movies","Action Movies","Horror Movies","Romantic Movies", "Electronic Engineering","Computer Engineering","Biology Engineering","Chemical Engineering", "Gothic Architecture","Baroque Architecture", "Modern Architecture","Renaissance Architecture", "Soccer","Basket","Baseball","Golf", "American Geography","Asian Geography","European Geography","African Geography"]
for topic in topics:
    seed = randint(0,1000)
Prompt
    messages = [
    {"role": "system", "content": "You are a helpful assistant. A topic with a random seed is given, please write a knowledge document within the topic."},
    {"role": "user", "content": f"topic:{topic},seed:{seed}"}
    ]
```

## C.2. Prompts for Generating the Initial Queries

To enhance the diversity of the initial queries while keeping them relevant for the victim documents, we design five different generation prompts and apply them in numeration for the casual-LLM. Content are following: To enhance the diversity of the initial queries while keeping them relevant for the victim documents, we design five different generation prompts and apply them in numeration for the casual-LLM. Content are following:

---

**Prompt 1**

messages = [ {"role": "system", "content": "You are a helpful query writer for information retrieval system. A knowledge document content is provided followed by the user. Please generate a query with content in which the human asker is facing a practical problem and the document would be helpful to solve the it."},
{"role": "user", "content": doc} ]

---

**Prompt 2**

messages = [ {"role": "system", "content": "You are a helpful query writer for information retrieval system. A knowledge document content is provided followed by the user. Please generate a query by questioning some factors of the document content."},
{"role": "user", "content": doc} ]

---

**Prompt 3**

messages = [{"role": "system", "content": "You are a helpful query writer for information retrieval system. A knowledge document content is provided followed by the user. Please generate a query targeting the document and containing five keywords from the document."},
{"role": "user", "content": doc} ]

---

**Prompt 4**

messages = [ {"role": "system", "content": "You are a helpful query writer for information retrieval system. A knowledge document content is provided followed by the user. Please generate a one-sentence summary for this document."},
{"role": "user", "content": doc} ]

---

**Prompt 5**

messages = [ {"role": "system", "content": "You are a helpful query writer for information retrieval system. A knowledge document content is provided followed by the user. Please generate a query that has similar semantics but contains few overlaps with the document."},
{"role": "user", "content": doc} ]

---

## C.3. Prompt for Poisoned-RAG

The following prompt is what we use for Poisoned-RAG attack:

---

messages = [ {"role": "system", "content": f"The user will followingly input a query and a document. Your task is to generate {num_sts} adversarial tokens following the end of the document and make the information retrieval system to rank the injected document lower for the given query. Please only output the adversarial tokens without any other explanation."},
{"role": "user", "content": f"Query: {query}"},
{"role": "user", "content": f"Document: {doc}"}
]

---

## D. Implementation and Dataset Details

The datasets we applied from (Hongjin et al.) is illustrated in the following tables: Table 8 describes the query amount,

| Dataset | $|\mathcal{Q}|$ | $|\mathcal{D}|$ | $|\mathcal{D}+|$ | $L_Q$ | $L_D$ |
|---|---|---|---|---|---|
| Econimics | 103 | 50,220 | 8.0 | 181.5 | 120.2 |
| Psychology | 101 | 52,835 | 7.3 | 149.6 | 118.2 |
| Biology | 103 | 57,359 | 3.6 | 115.2 | 83.6 |
| Robotics | 101 | 61,961 | 5.5 | 818.9 | 121.0 |

*Table 8.* Dataset Description

document amount, ground-truth document per query amount, average query length and average document length. In our experiment, we first attack every ground-truth document with injected tokens, and then retrieve the top 25/50 documents among the corpus by the query datas. Then we evaluate the Recall and NDCG based on how many ground-truth documents of the query are retrieved in this top samples and how they rank.

## E. Verifying $\epsilon$-$p_\epsilon$-Precise on Qwen, Jinaai, and Gemma

Table 9 shows the verification results (see Section 3.2) conducted on Embedding-Gemma-300M and Jinaai-Embeddings-v3. The $p_\epsilon$ of both are high enough to validate the truthfulness of the definition and deduction we made in Section 3.2.

| Topic Name | | Qwen1.5-7B-Ins. | | | | | | Embedding-Gemma-300M | | | | | | Jinaai-Embeddings-v3 | | | | | |
|---|---|---|---|---|---|---|---|---|---|---|---|---|---|---|---|---|---|---|---|
| | | $p_0$ | $p_{0.1}$ | $p_{0.2}$ | $p_{0.3}$ | In Sim. | Out Sim. | $p_0$ | $p_{0.1}$ | $p_{0.2}$ | $p_{0.3}$ | In Sim. | Out Sim. | $p_0$ | $p_{0.1}$ | $p_{0.2}$ | $p_{0.3}$ | In Sim. | Out Sim. |
| Engin. | Chemical | 100 | 99.6 | 91.7 | 87.8 | [0.67,0.90] | [-0.07,0.58] | 99.6 | 88.3 | 87.0 | 84.8 | [0.66,0.94] | [-0.01,0.70] | 100 | 100 | 99.6 | 88.3 | [0.82,0.95] | [-0.02,0.62] |
| | Computer | 92.6 | 87.0 | 87.0 | 66.5 | [0.49,0.87] | [-0.05,0.63] | 96.1 | 90.9 | 87.8 | 79.1 | [0.67,0.88] | [0.01,0.80] | 96.1 | 97.0 | 92.6 | 87.8 | [0.67,0.88] | [0.01,0.80] |
| | Electric | 92.6 | 89.6 | 83.0 | 47.8 | [0.50,0.89] | [-0.05,0.63] | 88.7 | 86.5 | 69.1 | 30.0 | [0.52,0.90] | [-0.00,0.80] | 100 | 97.4 | 90.0 | 87.4 | [0.75,0.94] | [-0.09,0.70] |
| | Biology | 99.1 | 93.9 | 88.3 | 86.1 | [0.55,0.85] | [-0.09,0.56] | 99.1 | 90.9 | 87.4 | 84.8 | [0.68,0.93] | [0.01,0.70] | 92.2 | 87.0 | 85.7 | 53.5 | [0.53,0.90] | [-0.02,0.62] |
| Movies | Action | 89.1 | 77.0 | 29.1 | 0.4 | [0.37,0.87] | [-0.06,0.60] | 97.4 | 90.4 | 29.1 | 20 | [0.52,0.92] | [0.00,0.63] | 100 | 100 | 97.0 | 90.4 | [0.73,0.90] | [-0.02,0.60] |
| | Horror | 100 | 94.3 | 90.9 | 82.2 | [0.59,0.88] | [-0.04,0.59] | 100 | 99.1 | 99.1 | 89.1 | [0.70,0.93] | [0.00,0.63] | 100 | 100 | 99.1 | 90.4 | [0.70,0.93] | [0.00,0.63] |
| | Romantic | 88.3 | 87.4 | 79.6 | 28.7 | [0.44,0.88] | [-0.05,0.63] | 100 | 96.5 | 79.6 | 55.2 | [0.59,0.91] | [0.01,0.58] | 99.6 | 91.3 | 87.4 | 79.1 | [0.56,0.92] | [-0.05,0.58] |
| | Comedy | 87.0 | 72.6 | 33.9 | 0.0 | [0.32,0.82] | [-0.07,0.63] | 81.7 | 53.9 | 33.9 | 0.0 | [0.36,0.87] | [-0.05,0.55] | 83.9 | 23.0 | 0.0 | 0.0 | [0.25,0.76] | [-0.09,0.58] |
| Achite. | Baroque | 99.6 | 94.3 | 91.3 | 87.4 | [0.70,0.94] | [-0.03,0.72] | 100.0 | 100 | 98.3 | 91.7 | [0.80,0.95] | [0.02,0.64] | 100 | 93.5 | 91.3 | 87.8 | [0.75,0.92] | [-0.02,0.71] |
| | Gothic | 100 | 96.1 | 91.3 | 87.0 | [0.71,0.91] | [-0.05,0.71] | 100 | 100 | 99.6 | 92.2 | [0.81,0.93] | [-0.02,0.62] | 99.6 | 95.2 | 90.9 | 86.5 | [0.71,0.94] | [-0.14,0.71] |
| | Renaissance | 100 | 90.9 | 87.0 | 87.0 | [0.64,0.89] | [-0.02,0.72] | 100 | 99.6 | 92.6 | 87.8 | [0.74,0.91] | [-0.00,0.64] | 100 | 97.8 | 91.3 | 87.0 | [0.76,0.90] | [-0.03,0.71] |
| | Modern | 100 | 100 | 87.8 | 86.5 | [0.65,0.88] | [-0.03,0.54] | 100 | 100 | 100 | 89.1 | [0.74,0.90] | [0.01,0.52] | 100 | 100 | 90.0 | 82.6 | [0.65,0.91] | [-0.07,0.54] |
| Geogra. | American | 84.8 | 78.7 | 50.4 | 5.2 | [0.40,0.86] | [-0.07,0.63] | 95.2 | 86.5 | 77.4 | 47.4 | [0.56,0.85] | [-0.03,0.62] | 99.6 | 88.7 | 81.7 | 68.7 | [0.61,0.92] | [-0.02,0.62] |
| | European | 92.2 | 87.0 | 86.5 | 78.3 | [0.57,0.89] | [-0.04,0.69] | 96.1 | 89.6 | 86.1 | 53.0 | [0.57,0.87] | [-0.03,0.65] | 99.6 | 90.0 | 80.0 | 50.0 | [0.60,0.91] | [-0.04,0.60] |
| | Asian | 89.1 | 87.0 | 87.0 | 80.0 | [0.54,0.85] | [-0.06,0.68] | 97.4 | 89.1 | 87.0 | 78.7 | [0.59,0.85] | [-0.04,0.66] | 100 | 96.5 | 87.4 | 83.5 | [0.68,0.93] | [-0.02,0.62] |
| | African | 91.3 | 87.4 | 87.0 | 83.0 | [0.58,0.85] | [-0.09,0.69] | 99.6 | 89.1 | 87.0 | 84.8 | [0.61,0.86] | [-0.05,0.66] | 100 | 91.7 | 87.8 | 80.9 | [0.61,0.92] | [-0.02,0.62] |
| Sports | Soccer | 100 | 97.8 | 97.8 | 91.7 | [0.71,0.89] | [-0.09,0.64] | 98.3 | 92.6 | 78.7 | 19.1 | [0.50,0.91] | [0.01,0.54] | 100 | 98.3 | 94.8 | 90.9 | [0.69,0.92] | [-0.02,0.61] |
| | Baseball | 100 | 93.0 | 91.7 | 91.7 | [0.66,0.92] | [-0.05,0.64] | 93.5 | 83.5 | 50.0 | 0.03 | [0.47,0.87] | [0.04,0.54] | 95.7 | 90.4 | 88.7 | 63.5 | [0.57,0.89] | [-0.14,0.61] |
| | Golf | 90.9 | 90.0 | 68.3 | 15.7 | [0.44,0.91] | [-0.08,0.63] | 100 | 98.3 | 87.4 | 49.6 | [0.60,0.90] | [-0.00,0.52] | 100 | 99.1 | 93.5 | 87.0 | [0.66,0.88] | [-0.05,0.60] |
| | Basketball | 87.8 | 82.6 | 13.9 | 0.0 | [0.31,0.59] | [-0.06,0.61] | 100 | 83.5 | 83.5 | 40.4 | [0.45,0.78] | [-0.05,0.43] | 92.6 | 58.7 | 16.5 | 0.0 | [0.36,0.82] | [-0.04,0.58] |
| Politics | American | 22.2 | 0.0 | 0.0 | 0.0 | [0.44,0.81] | [-0.04,0.62] | 88.3 | 67.8 | 37.4 | 0.05 | [0.47,0.91] | [-0.03,0.66] | 0.0 | 0.0 | 0.0 | 0.0 | [0.0,0.90] | [-0.05,0.61] |
| | Japanese | 92.2 | 88.3 | 80.4 | 36.5 | [0.44,0.81] | [-0.04,0.62] | 87.4 | 53.0 | 3.9 | 36.5 | [0.36,0.85] | [-0.05,0.66] | 97.4 | 89.1 | 87.0 | 78.3 | [0.59,0.91] | [-0.02,0.64] |
| | Chinese | 1.3 | 0.0 | 0.0 | 0.0 | [0.05,0.77] | [-0.06,0.55] | 0.07 | 0.0 | 0.0 | 0.0 | [0.16,0.92] | [0.01,0.66] | 90.0 | 86.5 | 73.5 | 46.1 | [0.51,0.89] | [-0.04,0.64] |
| | European | 76.5 | 30.9 | 1.7 | 0.0 | [0.28,0.87] | [-0.06,0.63] | 0.0 | 0.0 | 0.0 | 0.0 | [0.01,0.87] | [0.01,0.65] | 100 | 93.4 | 88.7 | 79.1 | [0.63,0.89] | [-0.01,0.61] |

*Table 9.* 24 context topics' $p_\epsilon(\%)$ on Qwen1.5-7B-Ins., Embedding-Gemma-300M, and Jinaai-Embeddings-v3 retrievers. "In Sim" describes the minimum and maximum of similarity between every pair within the topic corpus ($10\times10$), and "Out Sim" for every pair of one within the topic and one in other 23 topics ($10\times230$).

