# OpenReview forum: "``Someone Hid It!'': Query-Agnostic Black-Box Attacks on LLM-Based Retrieval"
_ICML.cc/2026/Conference — ICML 2026 regular_

### Official Review · Reviewer_mrTU · 2026-02-28

**Soundness:** 2
**Presentation:** 3
**Significance:** 3
**Originality:** 2
**Overall Recommendation:** 3
**Confidence:** 4

**Summary:**

This paper studies adversarial attacks on LLM-based retrieval (LLMR) under a query-agnostic black-box setting, where the attacker does not know the victim model, its document corpus, or user queries, and can only slightly modify a target document through limited token injection.

The authors observe a “topic clustering” phenomenon in embedding spaces and define an (\epsilon)-(p_\epsilon)-Precise property to describe retriever behavior. Based on this, they formulate a surrogate-based transfer attack as a min–max optimization problem over document–query similarity. They propose a Document–Queries Adversarial (DQ-A) learning method, which uses a surrogate LLM and another LLM to generate and adapt multiple query samples, aiming to learn transferable suffix tokens that reduce the document’s retrievability. They also experiment with using word-embedding-level representations to improve transferability.

Experiments on several benchmark datasets show retrieval performance drops (NDCG and Recall) across multiple open-source embedding models, compared with baselines such as GCG, PRADA, and Poisoned-RAG. Ablation studies analyze the effect of adversarial learning design and sampling size.

**Compliance With Llm Reviewing Policy:**

Affirmed.

**Final Justification:**

My main concern about Lemma 3.2 remains. The current response still does not rigorously justify why moving the attacked document outside the topic region under the surrogate model should imply a corresponding effect under the victim model. In my view, this gap is currently absorbed into the probability term rather than formally established. The added empirical verification is helpful, but it is only a relaxed approximation and does not validate the lemma as originally stated.

I decided to raise my score.

**Key Questions For Authors:**

1. Your experiments only evaluate single-stage embedding retrieval. In practice, many RAG systems use a two-stage pipeline (retriever + reranker). Have you tested the attack in such a setting? If a strong reranker (e.g., cross-encoder or BGE reranker) is applied, does the attack still reduce the final retrieval performance?

2. Would the conclusions change if other models, such as the BGE series(bge-m3, bge-reranker...), are evaluated?

3. The paper claims zero-shot transferability, but experiments are limited to embedding similarity models. Does this transferability extend to other architectures, such as hybrid retrieval systems or reranker-based pipelines?

4. Will the code and detailed implementation settings be released?

**Limitations:**

1. The experiments focus mainly on embedding-based retrievers and do not evaluate full RAG pipelines (e.g., with rerankers or production-level systems).

2.The theoretical discussion assumes topic clustering and cross-model alignment, but the practical stability of these assumptions is not deeply examined.

3. Since the method enables hiding documents from retrieval systems, it could be misused to suppress information or manipulate search visibility. The paper would benefit from a clearer discussion of safeguards, responsible disclosure, and possible defenses.

**Strengths And Weaknesses:**

Soundness

Strengths:
The paper studies a clearly defined and practically relevant threat model: query-agnostic, complete black-box attacks with limited suffix token injection.
The methodological flow is coherent: topic clustering observation (Fig.3), formalization via the ε–pε-Precise definition (Def.3.1, Eq.(2)), surrogate min–max objective (Eq.(6)), and the DQ-A adversarial learning approximation.
Experiments cover multiple BRIGHT subsets and several embedding models, evaluated with NDCG and Recall@25/50, and include ablations on sampling size and token length.

Weaknesses:

The topic clustering assumption is not uniformly supported. In Table 1, some topics (e.g., Politics-American/Chinese) show very low pε, which weakens the generality of the tight-cluster premise. The paper does not analyze how DQ-A performs in these weak-cluster cases.

The transferability argument (Lemma 3.2, Eq.(5)) implicitly assumes cross-model alignment between surrogate and victim embeddings, but no formal condition is provided to justify this step.

Baseline comparisons use different information settings (e.g., providing ground-truth queries and partial data access for some baselines), which makes the comparison less clean under a unified threat model.

Presentation

Strengths:
The paper is generally well organized, and the figures (especially Fig.4) help illustrate the DQ-A pipeline.
The experimental setup and evaluation metrics are clearly stated.

Weaknesses:

Some derivations, especially around Eq.(5), would benefit from clearer notation and more explicit assumptions.

Reproducibility could be improved by providing more detailed implementation settings, such as GCG search hyperparameters, the number of optimization steps, and details of token handling.

Significance

Strengths:
Security of LLM-based retrieval systems is an important and timely problem.
The query-agnostic black-box setting increases practical relevance compared to many prior white-box works.

Weaknesses:

All experiments focus on single-stage embedding retrieval. Modern RAG systems often use multi-stage pipelines with rerankers, so the demonstrated impact may mainly reflect first-stage vulnerability rather than end-to-end system behavior.

The robustness–precision trade-off observation (e.g., Qwen3-Emb-0.6B) is interesting but not deeply analyzed.

Originality

Strengths:
The strict query-agnostic black-box framing is meaningful.
The integration of surrogate transfer, adversarial document–query learning, and word-embedding-based surrogate signals is reasonably novel.

Weaknesses:

The method builds on existing token injection and surrogate transfer techniques; the main novelty lies in the combination and threat model rather than fundamentally new theory.

Transferability is only validated for embedding-based retrievers and not broader retrieval architectures.

---

> ### Author Rebuttal · Authors · 2026-03-31
>
> **We thanks the for recognition in our work’s significance and novelty, as well as the valuable suggestions.**
>
> **W1. DQA in weak-clusters**
>
> This weak-cluster case shows the other side of the coin: a weak-cluster also means a low precision of the retriever on the given topic, e.g., on Pol.-Chinese, most of other topic documents get competitive similarity with contexts inside the topic on Qwen-1.5. When user retrieves, a related document is likely to be out of retrieval itself, thus the attacker is likely in no need to attack. Besides, while in a weak-cluster the transferability is less promised, DQA is not definitely meant to fail empirically.
>
> **W2.L2. Cross-model alignment**
>
> We clarify that we **do not assume any alignment between surrogate and victim retrievers**, instead, we assume they have **independent outputs** on the same document. This is exactly why we study the topic-cluster property: since no direct similarity on models' output embeddings is promised, we **bridge them by their individual clustering behaviors for the same topic**. The estimated transferability only depends on both models’ $\epsilon-p^{\epsilon}$ precision on the topic requiring no direct relation.
>
> **W3. Different information setting**
>
> As we stated, DQA is the **first query-agnostic attack** on LLMR, it’s hard to directly apply baselines in the same setting as they all require victim queries. However, even giving baselines such a relaxed capacity, DQA still outperforms in the harder threat model, which further validates its practicality. Besides, in the ablation study we also test S-GCG that adapts GCG for the same setting with DQA.
>
> **W4. Eq.(5) Derivation**
>
> To better illustrate we will add a proof sketch. While due to the length limit we are unable to put it here, **please kindly check the proof in rebuttal to reviewer C2HN**
>
> **W5.Q4. Reproducibility details.**
>
> Thanks for concern on reproducibility. In **supplementary materials we have provided code** for verification. Our code will be opensourced upon the paper's acceptance.
>
> **W6.Q1.L1. Two stage pipeline.**
>
> This is a good question, and yes, in some scenarios the retriever is used with a reranker model: the retriever first picks a candidate pool and the reranker scores precisely within the pool. In this work we focus on the upper retrieval stage, aiming to **filter the victim document out of the pool** firsthand, and therefore **however strong the reranker is, the document is already out of retrieval**. To validate this, we test two retriever-reranker combinations. The retriever retrieves the top 25/50/100 documents first, filtering out others, and then the reranker scores retrieved documents and report the NDCG25/50/100. The first set is the suggested BGE-M3 retriever with BGE-reranker-v2-M3, and the second is Gemma-300M with Qwen3-reranker-0.6B:
>
> **Eco. N25  N50  N100|Psy. N25 N50 N100|Bio. N25 N50 N100|Rob. N25 N50 N100**
>
> **BGE-M3s**
>
> **Ori. |0.139 0.160 0.165|0.175 0.190 0.209|0.132 0.156 0.172|0.128 0.130 0.143**
>
> **Ours|0.131 0.151 0.160|0.166 0.179 0.197|0.121 0.145 0.164|0.109 0.122 0.135**
>
> **Gem.+Qwen3**
>
> **Ori. |0.212 0.230 0.263|0.236 0.264 0.276|0.249 0.280 0.323|0.175 0.188 0.211**
>
> **Ours|0.182 0.200 0.234|0.222 0.245 0.264|0.231 0.253 0.289|0.181 0.183 0.203**
>
> The results clearly show that in such pipeline DQA is still effective.
>
> **W7. Deeper trade-off analysis.**
>
> Thanks for interest in the analysis. By now we haven't found proper estimation way to explore it deeper. We’d like to try if recommended any approach.
>
> **W8. Novelty in the combination and threat model rather than new theory.**
>
> While appreciating the recognition on novelty of our threat model design, we still emphasize the differences of our method with existing works: while these works either simply assumes alignment in surrogate and victim models, or trains surrogate by interactions, we firstly investigate on (1)**``Why could transfer”**: theoretical analysis on necessary constraints to transfer an attack effectively, and (2)**``How to transfer”**: an accordingly designed framework to achieve such constraints. Our DQA is **fully motivated by methodological deductions** unlike existing empirically motivated methods, therefore, rather than a simple combination of existing works.
>
> **W9.Q3. Border models like BGE-M3**
>
> Here we test DQA on the raised BGE-M3 in default setting. The results help to prove DQA's effect on border models:
>
> **Data.|Eco. N25 N50 R25 R50|Psy. N25 N50 R25 R50|Bio. N25 N50 R25 R50|Rob. N25 N50 R25 R50**
>
> **Ori. |0.134 0.158 0.218 0.321|0.163 0.185 0.282 0.371|0.112 0.132 0.167 0.228|0.153 0.167 0.227 0.276**
>
> **Ours|0.127 0.151 0.205 0.313|0.150 0.171 0.277 0.360|0.101 0.120 0.158 0.215|0.137 0.156 0.201 0.270**
>
> **L3. Defense**
>
> We also discussed potential defense like **perplexity detection. Please kindly check our rebuttal to reviewer 5GJC**
>
> **We will add these clarifications and ablation studies in the final version, and provide all replication code in public.**

---

> > ### Author Rebuttal · Reviewer_mrTU · 2026-04-01
> >
> > Thank you for the authors’ response, which has addressed most of my concerns. However, I still have some concern about Lemma 3.2. In particular, I am not fully convinced by the second step: showing that the attacked document is pushed outside the topic region under the surrogate model does not necessarily mean it will also fall outside the corresponding region under the victim model. For example, the perturbation may mainly affect features that matter to the surrogate retriever, while the victim retriever may still keep the document close to the original query/topic.

---

> > > ### Author Response · Authors · 2026-04-01
> > >
> > > **We are happy to see that we have addressed most of the reviewer’s concerns, and glad to give further justification on the second deduction step!**
> > >
> > > This raised question is answered by the definition of our $\epsilon-p_{\epsilon}$ precision, which **exists a corresponding paired-value set on any given set X** for an embedding function f. Therefore, for the barrier on topic $\mathcal{X}$ we established in the surrogate model, there **always exist a corresponding $\epsilon-p_{\epsilon}$ for the exact same topic set $\mathcal{X}$**, and that’s the precision value that we care and use in our transferability guarantees. Therefore, **(1)** even at the same time **there also exist a potential set $\mathcal{X}’$** for victim model with slightly difference that **includes the rejected document $d’$** like the reviewer assumed, this set **just holds a different $\epsilon-p_{\epsilon}$ that we do not care about**. And **(2)** besides, since we assume **no relation** between the victim model and surrogate model, which **include both potential alignment and misalignment**, the raised potential error is simply accepted inside the distributional probability of the victim model— **as a regular case described in the $(1-p^{f}_{\epsilon})$ possibility**. This deduction way **tightly corresponds with our stated independence** on embedding functions, where we **do not need to guarantee for any specialized assumed misalignment**.
> > >
> > > ——————————————————————————————————————————————————————————————
> > >
> > >
> > > We find the reviewer **may still have confusion on the deduction process**. To help the reviewer for **better understanding how our deducted transferability works**, we **additionally provide a further empirical verification on the tranferability lemma** to show **its rationaility in practice**.
> > >
> > > **A."How we verify:"**
> > >
> > > To be specific, **for all tuples of query q, groundtruth document d and attacked d'** of in our main experiment results, we first **pick up the tuples that satisfy**:
> > > $$\max (\text{sim} (g(d'), g(q)), \text{sim} (g(d'), g(d))) \le \text{sim} (g(d), g(q))$$
> > > and **count how much fraction of these tuples achieve following for a given $\epsilon_f$ (chosen as 0, 0.05)**:
> > > $$\text{sim}(f(d'),f(q))\le \text{sim}(f(d),f(q)) - \epsilon_{f}$$
> > > This counted fraction provides **an approximating estimation for our  $p_{\epsilon}^{f}$** described in deductions.
> > >
> > > **B."Why could verify:"**
> > >
> > > Under this evaluation, we treat $d$ and $q$ as a **extremely small topic corpus $\mathcal{X}=${$q,d$}**. Noting such a weak corpus may not have a positive $\epsilon_{g}$ with $p_{\epsilon}^{g}=1$, without loss of generalization, we **simply give relaxation on the left hand** of the transferability with **removal of $\epsilon_{g} > 0$** and verify for:
> > > $$\max_{X\in\mathcal{X}} \text{sim}(g(d'),g(X)) \leq \min_{X_{i},X_{j}\in\mathcal{X}} \text{sim}(g(X_{i}),g(X_{j})) =  \text{sim}(g(q),g(d))\rightarrow^{p_{\epsilon}^{f}} \text{sim}(f(d'),f(q))\le   \min_{X_{i},X_{j}\in\mathcal{X}} \text{sim}(g(X_{i}),g(X_{j}))-\epsilon_{f} = \text{sim}(f(d),f(q)) - \epsilon_{f}  $$
> > >
> > > To verify the **practical ${p_{\epsilon}^{f}}$ probability** by expectation, we only need to count **how much fraction of tuples satisfying the left hand could also satisfy the righthand**, and that's exact **what we do in the above statment in A**.  We emphasize that **such relaxation on left hand does not reduce the strongness of verification, instead, posing more hardness**.
> > >
> > > **C.Verification results:**
> > >
> > > The evaluation results of each dataset are as follow:
> > >
> > > **Data. | Eco. | Psy. | Bio. | Rob.**
> > >
> > > **$p_{0}^{f}$|80.0%|69.7%|72.0%|58.9%**
> > >
> > > **$p_{0.05}^{f}$|7.6%| 3.7%| 0.9%|1.4%**
> > >
> > > Given the **relatively small topic corpus** constructed with **low $\epsilon-p^{f}_{\epsilon}$** and **hardness brought by left hand relaxation**, our **considerable verifed probability on $\epsilon_{f}=0$** and **non-zero probability on on $\epsilon_{f}=0.05$** could **partially prove the rationality of our theoriticial deduction steps and conclusion**.
> > >
> > > **We hope this approximating toy-verification could help you understand why our deduction could work.**
> > >
> > > ——————————————————————————————————————————————————————————————
> > >
> > >
> > > **We sincerely hope the reviewer could carefully consider the justification we raised, and re-evaluate on the review recommendation with all concerns we have effectively addressed!**
> > >
> > > ——————————————————————————————————————————————————————————————
> > >
> > > **Thanks for your further justification and increase on the score!** We will **update all discussed clarifications**, **repharse relevant claims** and **provide a more detailed emprical verification** to give a clearer and convincing presentation in our next/final version.

---

### Official Review · Reviewer_5GJC · 2026-03-06

**Soundness:** 2
**Presentation:** 2
**Significance:** 2
**Originality:** 3
**Overall Recommendation:** 4
**Confidence:** 3

**Summary:**

This paper proposes a black-box attack for LLM-based retrieval systems that generates transferable adversarial tokens using zero-shot surrogate LLMs to promote or suppress documents during retrieval. The authors introduce a theoretical framework to analyze the factors, including victim document embeddings and the precision of surrogate and target models. They then optimize adversarial tokens through adversarial learning. The method further exploits query population imbalance and word-embedding surrogates to improve attack effectiveness.

**Compliance With Llm Reviewing Policy:**

Affirmed.

**Final Justification:**

The response has largely resolved my primary concerns, and I have raised my score to 4.

**Key Questions For Authors:**

1. How are the topics in Table 1 constructed, and how do you ensure the semantic consistency of the generated samples?
2. Why are Recall@25/50 and nDCG@25/50 used instead of nDCG@10, which is the standard metric in BRIGHT? Can the authors report nDCG@10 and Recall@10?
3. Which model was used in experiments: Qwen-1.5-7B or Qwen-1.5-7B-Instruct?
4. Can the attack be evaluated on newer models, such as Qwen2.5, to better understand robustness across model generations?
5. Why does the attack perform weakly on Qwen3-Embedding-0.6B, and what insights can be drawn from this observation?
6. Can the authors provide examples of adversarial documents and analyze whether they can be detected using simple defenses such as perplexity filtering?

**Limitations:**

1. Evaluation is conducted on only one dataset.
2. The evaluation metrics differ from the original benchmark.

**Strengths And Weaknesses:**

## Strengths

1. The paper studies black-box attacks on LLM-based retrieval systems using transferable adversarial tokens, which is a relevant and timely problem.
2. The theoretical formulation connecting surrogate and target model precision provides a reasonable perspective for analyzing transferability.
3. Experiments cover different model sizes and architectures, which is valuable for understanding attack behavior.

---

## Weaknesses

1. It is unclear how topics in Table 1 were selected and how the generated samples are verified to belong to the corresponding topic semantically.

2. The paper assumes that topics form tight embedding clusters, but Table 1 shows that this does not always hold. The relationship between topic clusters and the probability \(p_e\) is unclear, and it is not obvious whether the proposed definition holds across all topics.

3. Experiments are conducted on a single dataset, which makes it difficult to assess the robustness and generalization of the proposed attack.

4. The paper evaluates on the BRIGHT dataset but reports Recall@25/50 and nDCG@25/50, whereas the original benchmark reports nDCG@10. The paper should justify this choice and report results using nDCG@10 and Recall@10 for fair comparison.

5. Section 4.1 mentions Qwen-1.5-7B, while the table reports Qwen-1.5-7 B-Instruct. It is unclear which model was used.

6. The attack seems stronger on larger or older models and weaker on smaller or newer models, but the paper does not analyze this phenomenon. Evaluating newer models (e.g., Qwen2.5) would help to understand the result further. In addition, the weak performance on models such as Qwen3-Embedding-0.6B is not discussed.

7. The similarity constraint uses \( \delta \), the length of injected tokens, which does not necessarily reflect semantic similarity between \(d\) and \(d'\). For instance, a short sequence of random tokens could drastically reduce similarity, while a long but coherent sentence may preserve semantic similarity. The constraint, therefore, does not adequately capture the intended notion of similarity.

8. Since the optimization method (GCG) does not include perplexity regularization, the generated adversarial documents may have high perplexity and could be filtered by simple defenses. The paper does not analyze this.

9. Several figures are difficult to read and should be improved.

---

> ### Author Rebuttal · Authors · 2026-03-31
>
> **We thanks for the recognition of the established transferability analysis and the problem importance, as well as the suggestions on presentation and experiments**
>
> **W1.Q1. Topics selection and samples generation**
>
> The topics are simply chosen by human-crafting, ensured to present a hierarchy of big domains (e.g., movies) and small categories (e.g., comedy). The samples are generated by prompts shown in Appendix, and we have verified them in human. We will make the samples public for reader verification in the final code base.
>
> **W2. Cluster property not hold**
>
> We clarify that **there’s no case** that our defined $\epsilon-p_{\epsilon}$ precision “not hold” for a topic. The $p_{\epsilon}$ is a probabilistic constant, which evaluates the distributional probability of *an embedding from other topics holds a $\epsilon$ similarity gap*. Even when the topic embeddings are totally loose, the $p_{\epsilon}$ just drops to 0, only indicating the property is weak and the transferability hard to be guaranteed. Besides, while there exist a few cases that our transferability is not well-guaranteed, it doesn't mean our attack will definitely fail empirically.
>
> **W3. A new dataset**
>
> Here we include a new dataset "MS-500", which has 500 queries and 500K documents randomly sampled from *Msmarco-Passage*[1]. Due to length limit here we only show results of our attack on gemma, where it keeps effective:
>
> **Metric| N25  N50 R25 R50**
>
> **Orig.| 0.681 0.686 0.872 0.897**
>
> **Ours| 0.645  0.649 0.845 0.886**
>
> We will add full results in the final version.
>
> **W4.Q2. Justify metrics**
>
> We report Recall@25/50 and NDCG@25/50 since our attack is against **retrieval task**, which roughly filters documents to a smaller pool firstly, and in some scenarios may further forward to a reranker stage. Thus, **scales of 25 and 50 fit our case better**. Results of 10 follow a similar decrease trend as 25\50, while due to space limit we omit it here.
>
> **W5 Q3.** Thanks for raising the misuse, we refers to Qwen-1.5-7B-Instruct.
>
> **W6.Q5. New\small models**
>
> First we clarify that for “newer” models like **Embedding-Gemma-300M(2025 Sept)** and “smaller” models like **Jinaai-v3**, our attack performs the best against all baselines. We also **have given an analysis on the weak attack performance on Qwen3-embedding-0.6B, summarized as a trade-off** between the model’s original accuracy and robustness. This corresponds to the intuition on our transferability: *the stronger the victim model is, the more transferable our attack is*.
>
> **W7. Similarity**
>
> The similarity constraint simply means **content overlap** between the original and the attacked documents, following typical adversarial attack settings. As the similarity is higher, perturbations are fewer and the two documents looks more similarly, making it harder to be classified by users. Noticing this definition may be confusing we will replace it with “attacker capacity”.
>
> **W8.Q6. Perplexity detection.**
>
> This question is interesting and worth discussion. Here we conduct an evaluation on perplexity of our attacked documents, and compare with the original ones. Specifically, we compare with (1) corresponding original document perplexity, (2) the 100th largest and (3) the largest perplexity of the original corpus (each dataset 50K around docs). The results are followed:
>
> **Compared|Eco.| Psy. | Bio. |Rob.**
>
> **Original|100%|100%|100%|100%**
>
> **100th L.|7.3% |3.5% |11.4%|2.7%**
>
> **Largest| 0.0% |0.0%| 0.0% |0.0%**
>
> We first notice that **indeed our attack increases all victim documents perplexity**. However, comparing with the whole original corpus, **most of our attacked documents do not override the highest perplexity part (top 100), and none of them exceeds the highest one**. This suggests our attacked documents are still **in the original corpus’ natural perplexity range without outstanding gap**. Noting our attack is on a single document rather than creating a set of documents, it is hard to be detected: to detect the attacked document by perplexity, even if the detector blindly removes top 100 perplexity documents, ours is still likely to survive. Therefore, **the perplexity detection is not effective for our attack.**
>
> For the attacked example, here we post a shortened version:
>
> *Evidence[edit] Growth and aging[edit] There is a large body of evidence  …(7012 characters except space) …through clearance and recycling of damaged proteins and organelles..basicConfig useContext.dispatchEvent ArgumentOutOfRangeException corres SNMP sede masturb-Javadoc lineno*
>
> This attacked example is from Bio. dataset. Compared with the original long context, our injected tokens are **short and have no outstanding characters**.
>
> **W9: Figures.** We will improve them.
>
> **We will add these clarifications and full experiments in the final version, and provide corresponding replications in our public code.**
>
> [1] MS MARCO: A Human Generated MAchine Reading COmprehension Dataset, Payal Bajaj et al., 2018.

---

> > ### Author Rebuttal · Reviewer_5GJC · 2026-04-03
> >
> > Thank you for the detailed rebuttal and additional experiments. The response has largely resolved my primary concerns, and I have raised my score to 4.

---

> > > ### Author Response · Authors · 2026-04-03
> > >
> > > **We are glad to know that we have addressed your concerns, and sincerely thanks for your recoginition on our work and effort!**

---

### Official Review · Reviewer_C2HN · 2026-03-12

**Soundness:** 2
**Presentation:** 3
**Significance:** 2
**Originality:** 3
**Overall Recommendation:** 3
**Confidence:** 3

**Summary:**

This paper proposes a query-agnostic document-hiding attack on LLM retrieval systems, which aims to make a target document less likely to be retrieved. The paper considers a black box setting that the attack is generated without knowledge of real user queries or access to the target retriever. Experiments on multiple dense retrievers show consistent drops in recall, demonstrating the success of attack.

**Compliance With Llm Reviewing Policy:**

Affirmed.

**Ethical Review Concerns:**

No concerns

**Key Questions For Authors:**

1. Could you provide a formal proof of Lemma 3.2? Also, what's the meaning of $\overset{p_{\varepsilon}^{f}}{\rightarrow}$ in equation (5)? This is not a standard notation and perhaps needs more clarification.

2. The proposed method, DQ-A learning, still needs to use GCG for generating suffix tokens, which may be computationally expensive at scale. Could you discuss its computational cost and practicality?

3. What's the definition of $dis()$ in equation (1)?

4. In lines 303 - 304, "for results not lower than 0.5%, we mark them in gray, indicating insufficient attacks", is there any justification for setting this threshold?

5. In line 146, should "document corrupt D" be "document corpus D"?

**Limitations:**

yes

**Strengths And Weaknesses:**

* Strength:

The proposed attack is query-agnostic. It does not require access to the document corpus. which makes the threat model more realistic and broadly applicable.

* Weakness:

The theoretical contribution and its analysis: establishing a formal framework for the attack claimed by the paper relies heavily on the topic-wise clustering hypothesis and several heuristic approximations, which may need further justification.

---

> ### Author Rebuttal · Authors · 2026-03-31
>
> **We are thankful for the recognition of the practicality of our attack and valuable suggestions to refine our paper!**
>
> **Q1: A formal proof for Lemma 3.2.**
>
> Thanks for raising this issue. Here we provide a proof sketch for our Lemma 3.2 below, please kindly check it. Besides for the $→^{p_{\epsilon}^{f}}$, it means “in possibility of $p^{f}_{\epsilon}$”.
>
> **Proof sketch.** On the surrogate model $g$, when the maximal similarity of the injected document $d’$ with the topic cluster set is lower than the minimal similarity within the topic cluster with a positive gap value $\epsilon_{g}$, we could infer the document is injected outside the set :
> $$(\max_{X\in \mathcal{X}} \text{sim}(g(d’),g(X))\leq \min_{X_{i},X_{j}\in \mathcal{X}} \text{sim}(g(X_{i}),g(X_{j}))-\epsilon_{g}) \wedge (\epsilon_{g}>0) → d’\notin \mathcal{X}$$
> Since the victim model $f$ is black-box to us, we can only assume it as an independent embedding function with the surrogate model. However, as the victim model $f$ follows the $\epsilon_{f}$-$p_{\epsilon}^{f}$ precision on this cluster set $\mathcal{X}$, which describes a probabilistic for documents on non-relevant distribution , we could simply estimate outside d’ document a probabilistic bound by this property:
> $$d’\notin \mathcal{X} →^{p_{\epsilon}^{f}} \text{sim}(f(d’),f(q)) \leq \max_{X\in \mathcal{X}} \text{sim}(f(d’),f(X))\leq \max_{X_{i},X_{j}\in \mathcal{X}} \text{sim}(f(X_{i}),f(X_{j}))-\epsilon_{f} $$
> This two-step deduction proves our lemma.
>
> **Q2: Computational cost and practicality.**
>
> Here we provide the time complexity of our attack and empirical time cost in average of our method. $V$ and $D_{dim}$ refers to the vocabulary size and the dimension size of the surrogate LLM, $|S|$ refers to the sample queries amount, $|d|$ refers to the victim document length and $\delta$ refers to amount of the tokens we inject. Since our attack aims to decrease the retrieved possibility of **a specific victim document instead of creating batches of corruption documents**, this attack computation cost is acceptable.
>
> **Time Complexity Per Epoch                                           | Average Time Per Document**
>
> **$O(|S|(|d|+\delta)^{2}D_{dim} + \delta V$)         |      23.065s (10 injected tokens, 20 epoches)**
>
> **Q3: The definition of distance**
>
> Sorry for not clarifying this clean enough. This $dis$ means the content distance between the injected document and the original ones, which is indeed the amount of tokens we injected.
>
> **Q4: The meaning of “0.5% as grey”.**
>
> This grey simply means the performance decrease (compared with the original performance) is less than 0.5%, which indicates the attack performance is not effective enough. For ease of expression we mark their column as grey.
>
> **Q5: Miss use of the word.**
>
> Thanks for the notice! We’ll fix it.
>
> **In all, we will add the proof sketch, complexity analysis and these necessary clarifications into the final version of our paper.**

---

> > ### Author Rebuttal · Reviewer_C2HN · 2026-04-03
> >
> > Thanks for your response. Some of my questions have not been fully resolved.
> >
> > * Q1
> >
> > What does “in possibility” mean? Do you mean under the randomness of ...?
> >
> > What does $\rightarrow$ mean here? Does it mean “converges to” or “implies”? I am quite confused about this part.
> >
> > * Q3
> >
> > Could you provide a formal definition of $dis$? What is its formula?
> >
> >
> > * Q4
> >
> > I think my question hasn't been resolved. What is the criterion for choosing this threshold?
> >
> > -------------------------Update-----------------------
> >
> > Thanks for your response. The notation is clear to me now. However, I still have concerns about the paper’s theoretical contribution, which currently feels somewhat overstated. Rephrasing the relevant claims and providing a clearer explanation of the mathematical notation would improve the paper’s overall quality. Overall, I will maintain my score and recommend a borderline accept.

---

> > > ### Author Response · Authors · 2026-04-03
> > >
> > > **Thanks for your feedback, we'd like to give further justification on your remaining questions!**
> > >
> > > **Q1. Meaning of "In possibility" and $\rightarrow$**
> > >
> > > *In possibility of $p_{\epsilon}$* means, **"there's probability of $p_{\epsilon}$ that"**, which does describe a **statistical randomness**. $\rightarrow$ does mean **"implies"** here. Combining them together, the expression we use $A \rightarrow^{p} B$ means **"if A happpens, then there's probability of $p$ that B will happen"**. It seems like we have a misuse of word "possbility" with "probability", we will replace it.
> > >
> > > **Q2. dis function formulation.**
> > >
> > > The **dis** function means the "data distance" which is widely adopted in adversarial attack settings. In this paper we simply refer to the **injected tokens applied to the document, which could be formally expressed as:**
> > > $$dis(A, B) =|B| - |A| \quad  \text{if } A \sqsubseteq B;\quad \infty \quad \text{otherwise}$$
> > >
> > > We note $A \sqsubseteq B$ means $A$ is a subsequence of $B$ (meaning all tokens of $A$ appear in $B$ in their original relative order). Considering the confusion potentially brought to the readers, we will replace the "**dis**" function name with "**injection budget**" for better expression in our final version and give the above formulation for explaining.
> > >
> > >
> > > **Q4. Criterion for choosing 0.5%**
> > >
> > > We further clarify this *less than 0.5%* **does not play any role as a threshold, criterion or hyperparameter** in our method, but just a pre-set vaule for **better table presentation deciding the marked color**. The value of 0.5% is a relatively small value that we choose in human sense, which indicating the transfered attack is **nearly in no effect on direct observation**. Considering this presentation confusion that may bring, **we will remove this color marking standard in presenting the table and change related result items back to the normal color**.
> > >
> > > **We hope our response could address your remaining concerns on the presentation and look forward to your further feedback!**
> > >
> > > ————————————————————————————————————————————————————
> > >
> > > **Thanks for your further feedback and support on boarderline accept!** We will **update all discussed notation clarifications**, **rephrase the relevant claims** and **refine for a clearer proof dedcution** as suggested to make the theorerical contribution in a more precise presentation in our next/final version.

---

### Official Review · Reviewer_C8kX · 2026-03-13

**Soundness:** 3
**Presentation:** 2
**Significance:** 3
**Originality:** 2
**Overall Recommendation:** 4
**Confidence:** 4

**Summary:**

This paper introduces a query-agnostic, fully black-box adversarial attack framework targeting Large Language Model-based Retrieval  systems. It addresses a critical gap in existing research, where most attacks rely on unrealistic assumptions such as prior knowledge of user queries or white-box access to the model. The paper proposes a "Query-Document Adversarial Learning" mechanism, utilizing a third-party generative model to sample a diverse set of potential queries and optimizes a small set of adversarial tokens to push the document's semantic representation away from its original topic cluster.

**Compliance With Llm Reviewing Policy:**

Affirmed.

**Final Justification:**

The paper proposes a novel method for adversarial attacks on retrieval systems, with solid empirical results. And most of my concerns are addressed during the rebuttal process. I keep my score unchanged.

**Key Questions For Authors:**

1. In Section 6, the paper discusses the potential for extending the framework to boost the ranking of specific documents. Could the authors provide experimental evidence or quantitative results demonstrating the improvement in retrieval rank when the adversarial learning loss is reversed.

**Limitations:**

yes

**Strengths And Weaknesses:**

Strengths:
1. By focusing on a fully black-box and query-agnostic scenario, the paper closely mirrors real-world attack. This significantly enhances the study's value for the industry when designing defenses against adversarial threats.
2. The proposed "Query-Document Adversarial Learning" algorithm is both novel and effective. The integration of LLM-driven query diversification with a surrogate embedding layer optimization strategy allows the attack to achieve high performance across multiple state-of-the-art retrievers

Weaknesses:
1. The process of identifying optimal adversarial tokens requires extensive model inference and backpropagation. For attackers targeting large-scale databases, the required computational cost may be prohibitive, which could limit the feasibility of the method as a tool for large-scale disruption.
2. The theoretical foundation (e.g., $\epsilon-p_\epsilon$-PreciseRetriever) assumes that document topics are clearly defined. However, in practice, many documents are multidisciplinary or cover overlapping topics. In these fuzzy semantic boundary regions, it may be considerably more difficult to push a document away from all relevant clusters using only a few tokens, potentially reducing the attack's success rate.
3. The attack is currently limited to suffix-only injections. Real-world systems often employ truncation (e.g., only reading the first 512 tokens) or specialized pooling strategies for long-form documents. The effectiveness of the attack might be significantly compromised under such common deployment configurations.

---

> ### Author Rebuttal · Authors · 2026-03-31
>
> **We thanks for the reviewer positive feedback and recognition, and appreciate the constructive suggestions on validating and improving the capacity of our work!**
>
> **W1: Targeting large-scale databases, the computational cost may be large.**
>
> Thanks for raising this scalability problem, which also helps us to clarify the scope between our method and corruption attacks: as we stated, our attack is defined as **against one specific document regardless of the remaining document corpus**. Therefore our complexity only relies on (1) the victim document length and (2) the surrogate model’s scales. This computation invariance on corpus suggests our specific attack has advantage on scalability compared with traditional corruption attacks.
>
> **W2: Cross-topic documents can be vague to be attacked.**
>
> We’d like to clarify that **“cross-topic”** could simply be viewed as **an independent topic** in our defined framework. Even if there is slight overlap between the original topics and the cross topic distribution, since our “similarity gap” is described as a probabilistic gap under the whole document corpus distribution, this small overlap could be accepted into our framework and won’t hurt on the transferability. To better illustrate how our theoretical transferability would behave on such cross-topic documents, we further generate 10 documents for each cross topic of *“Asian Politics Related to China and Japan”*, *“Comedical Romantic Movies”*, *"Electronic and Computer Engineering"* and *"Gothic and Baroque Architecture"*, and respectively estimate their $p_{\epsilon}$ when mixing with the previous generated 240 documents in Table 1. The results are follow:
>
> **$p_{\epsilon}$ |  CJP  | CRM  |  ECE  | GBA**
>
> **$p_{0}$ | 0.98 | 0.94 | 0.91 | 1.00**
>
> **$p_{0.1}$| 0.92| 0.93 | 0.85 | 0.92**
>
> **$p_{0.2}$| 0.83| 0.90 | 0.84 | 0.92**
>
> **$p_{0.3}$| 0.59| 0.81 | 0.68 | 0.89**
>
> We could see that even when comparing with document corpus containing “vague” topics, e.g., "Gothic Architecture" to "Gothic and Baroque Architecture", there still exists **a good probabilistic similarity gap between the clusters**. Therefore, the “cross-topic”, which in human sense may exhibit as a mixed type compared with “normal topic”, indeed empirically behaves as an independent topic with a distinct cluster like a “normal” one,  and therefore there’s no worry on such vagueness or potential influence on attack performance.
>
> **W3: Specialized preprocess may make attack insufficient.**
>
> This is an interesting question deserving exploration. Here we first provide **an ablation study** below on the **injected token positions**: Suffix refers to the attack we apply in the main experiments; in Prefix we change the injected position to the beginning before the first word of the document; and in Middle we change the injected position to the $\frac{|d|}{2}$ location of the document. While at the middle position our attack effect may be slightly weaker than perfix or suffix, **attack at all positions still show considerable decrease compared with the original performance**. Secondly, we also glad to mention that for retrievers we use in the experimental section, there exist models like **jinaai and gemma which already have appled preprocesses like truncation** before aggregating the final embedding, and still not able to prevent our decrease effects. Therefore **such specialized preprocess can not pose as a barrier against our attack**.
>
> **Pos. | Eco. N25 N50 R25 R50  | Psy. N25 N50 R25 R50 | Bio. N25 N50 R25 R50 | Rob. N25 N50 R25 R50**
>
> **Ori. |0.225 0.245 0.307 0.395|0.252 0.285 0.395 0.523|0.223 0.254 0.313 0.420|0.178 0.197 0.262 0.336**
>
> **Pre. |0.196 0.216 0.286 0.366|0.225 0.246 0.373 0.453|0.196 0.219 0.302 0.383|0.162 0.187 0.235 0.334**
>
> **Mid.|0.210 0.232 0.288 0.381|0.249 0.277 0.401 0.502|0.214 0.247 0.316 0.438|0.163 0.186 0.242 0.332**
>
> **Suf. |0.191 0.214 0.278 0.374|0.229 0.259 0.363 0.486|0.209 0.238 0.300 0.402|0.160 0.182 0.251 0.334**
>
> **Q1: Experimental evidence for reversed adversarial loss increases document retrieval.**
>
> Yes, we have done a small test as follow: we first randomly pick a document from the original economics dataset's corpus, and change all queries’ ground truth to be only the picked document, which results in a nearly zero performance, with Recall50 only 0.00227 and NDCG50 only 0.00116. Then we reverse the losses of our attack, change the maximum term respect to sampled queries to minimum terms, and apply the reversed attack to the picked document. The Recall50 is then raised to  0.00601 and NDCG50 to 0.00251. Though this **doubled effect** is still limited in lack of further refining, it shows that **it's possible to use our attack for enhancing the document retrieval**, and we will improve and explore this reversed method further in the future work.
>
> **In all, we will add all these clarifications and ablation analysis in the final version, and provide corresponding replications in our final public code**.

---

> > ### Author Rebuttal · Reviewer_C8kX · 2026-04-03
> >
> > Thanks for the response. Most of my concerns are addressed. I'll keep my score unchanged.

---

> > > ### Author Response · Authors · 2026-04-03
> > >
> > > **We are glad to know that we have addressed your concerns! Thanks for your recognition on our work!**

---

### Decision · Program_Chairs · 2026-04-30

**Decision:**

Accept (regular)

**Comment:**

This paper studies adversarial token injection attacks on dense retrieval systems and proposes a framework grounded in a topic-clustering assumption. The authors also consider an optimization procedure to generate adversarial suffixes. The average score of this paper falls into a borderline case, but **from different reviewers, there is general agreement that the paper is interesting and potentially valuable, and this is indicated in the post-rebuttal scores (mostly weak accept).** The rebuttal addressed several concerns, including clarification of notation, additional experiments, and improved empirical validation.
Despite some concerns remaining, this paper presents critical empirical studies and relevant attack settings, and the rebuttal has addressed a substantial portion of the reviewers’ questions. While the theoretical justification remains incomplete and some assumptions are stronger than what is validated, the work is still considered meaningful for the community, particularly from an empirical and security perspective. Overall, after rebuttal, the reviewers find the empirical results promising and acknowledge the potential impact of the proposed attack formulation, and the AC would like to vote to accept this paper.